# Fasting induces metabolic switches and spatial redistributions of lipid processing and neuronal interactions in tanycytes

Maxime Brunner[1,2,6,7], David Lopez-Rodriguez [2,3,6,7], Judith Estrada-Meza [2,3], Rafik Dali[2,3], Antoine Rohrbach[2,3], Tamara Deglise[2,3], Andrea Messina [1,2], Bernard Thorens [4], Federico Santoni [1,2,5,7] ✉ & Fanny Langlet [2,3,7] ✉

The ependyma lining the third ventricle (3V) in the mediobasal hypothalamus plays a crucial role in energy balance and glucose homeostasis. It is characterized by a high functional heterogeneity and plasticity, but the underlying molecular mechanisms governing its features are not fully understood. Here, 5481 hypothalamic ependymocytes were cataloged using FACS-assisted scRNAseq from fed, 12h-fasted, and 24h-fasted adult male mice. With standard clustering analysis, typical ependymal cells and β2-tanycytes appear sharply defined, but other subpopulations, β1- and α-tanycytes, display fuzzy boundaries with few or no specific markers. Pseudospatial approaches, based on the 3V neuroanatomical distribution, enable the identification of specific versus shared tanycyte markers and subgroup-specific versus general tanycyte functions. We show that fasting dynamically shifts gene expression patterns along the 3V, leading to a spatial redistribution of cell type-specific responses. Altogether, we show that changes in energy status induce metabolic and functional switches in tanycyte subpopulations, providing insights into molecular and functional diversity and plasticity within the tanycyte population.

The central nervous system comprises numerous neural cells with a high morphological, molecular, and functional diversity, interconnected in precise and dynamic networks to ensure countless physiological functions. Among these functions, energy balance is mainly orchestrated by hypothalamic and brainstem circuits[1,2]. Neurons and glial cells, including astrocytes, tanycytes, and microglia, compose these circuits and regulate food intake, energy expenditure, and glucose homeostasis[3–5].

Among glial cells, ependymocytes lining the third ventricle (3V) are now recognized as full-fledged actors in regulating energy balance and glucose homeostasis[6,7]. The striking feature of the ependyma in

the mediobasal hypothalamus (MBH) is its high heterogeneity, conferring various biological functions[8]. First, lining the top of the 3V, typical ependymal cells are cuboid and ciliated-epithelial glial cells that control cerebrospinal fluid (CSF) homeostasis and waste clearance[9]. A coordinated ciliary beating of these cuboidal cells allows the maintenance of CSF flows through the ventricular system[10]. Interestingly, the metabolic peptide melanin-concentrating hormone increases cilia beat frequency, increasing CSF flow and volume transmission to possibly meet metabolic needs[11]. Besides typical ependymal cells, tanycytes are polarized ependymocytes lining the bottom and the lateral walls of the 3V in the tuberal region of the hypothalamus[7,12]. Their

[1]Service of Endocrinology, Diabetology, and Metabolism, Lausanne University Hospital, Lausanne, Switzerland. [2]Faculty of Biology and Medicine, University of Lausanne, Lausanne, Switzerland. [3]Department of Biomedical Sciences, Faculty of Biology and Medicine, University of Lausanne, Lausanne, Switzerland. [4]Center for Integrative Genomics, Faculty of Biology and Medicine, University of Lausanne, Lausanne, Switzerland. [5]Institute for Genetic and Biomedical Research (IRGB) - CNR, Monserrato, Italy. [6]These authors contributed equally: Maxime Brunner, David Lopez-Rodriguez. [7]These authors jointly supervised this work: Federico Santoni, Fanny Langlet. ✉e-mail: federico.santoni@chuv.ch; fanny.langlet@unil.ch

unique morphology and location allow them to form a triple interface crucial for regulating energy balance[13–15]. Indeed, their cell bodies lining the ventricular wall allow the sensing of the central metabolic state through the CSF[16,17]. Their long cytoplasmic extensions into key hypothalamic brain areas -including the median eminence (ME), the arcuate nucleus (ARH), the ventromedial nucleus (VMH), and the dorsomedial nucleus (DMH)- enable the integration into circuits regulating energy balance[15]. Finally, their endfeet contacting vessels in these brain areas allow the reception of metabolic information from the bloodstream[18–20]. Besides this strategic location, tanycytes are also highly heterogeneous, with distinct gene expression, morphological, and functional properties[7,8]. Notably, in the context of energy balance regulation, tanycytes have been described as regulators of blood-brain[18,21] and blood-CSF[19,22,23] exchanges, controllers of neurosecretion into the peripheral circulation[24], transporters and sensors of peripheral metabolic cues[16,19,23,25,26], coordinators of neuronal functions[17,27,28], and modulators of neural circuits through their neural stem cell properties[29–31]. Thus, based on their location along the 3V and these functional disparities, tanycytes are historically classified into four subtypes: β1, β2, α1, and α2[32,33]. β2-tanycytes line the floor of the 3V in the ME, contact the fenestrated vessels of the hypothalamic-hypophyseal portal system, and mainly regulate blood-brain exchanges. β1-tanycytes line the lateral evaginations of the infundibular recess and the ventromedial arcuate nucleus (vmARH), contact *en-passant* different neural cells including neurons, and end on the lateral ME fenestrated vessels or the pial surface of the brain to control neurosecretion. α2- and α1-tanycytes line the dorsomedial arcuate nucleus (dmARH) and the VMH/DMH respectively, contact different neural cells before ending on blood-brain barrier vessels, and are mainly described as fuel sensors and neuronal modulators.

Although this tanycyte classification has been widely used in literature, recent advances in our understanding of ependyma physiology revealed that it is no longer adequate[7,8]. Indeed, although marker genes are used to roughly separate tanycyte subtypes and typical ependymal cells, many genes exhibited a gradual pattern along the 3V rather than a clear-cut distribution across subpopulations[8]. Additionally, tanycytes lining the vmARH display morphological and functional features of α or β subgroups depending on the metabolic state, revealing complex and dynamic heterogeneity[18]. Therefore, uncovering this complexity is crucial for apprehending tanycytes' role and functional plasticity in metabolism.

In this study, we used fluorescence-activated cell sorting (FACS) associated with single-cell RNA sequencing (scRNAseq) and complementary analytic approaches to profile the typical ependymal cells' and tanycytes' transcriptional signatures and elucidate the dynamical changes resulting from an energy imbalance in adult male mice. First, using clustering analysis, we confirmed that our current tanycyte classification is inadequate as numerous gene markers are shared between subpopulations, and their heterogeneity further increases with a fasting time course. To overtake this, we developed a pseudospatial and temporal analysis to elucidate the ependyma neuroanatomical-like heterogeneity across the fed→fasting conditions. This approach allowed us to distinguish specific versus shared markers among the ependymal subpopulations, uncover high dynamics for the subgroup facing the ARH-VMH, and validate undiscovered functions for tanycyte populations involved in energy balance regulation.

## Results
### Transcriptomic profiling of ependymocytes along the 3V
To analyze the molecular signature of MBH ependymocytes, we performed FACS-associated scRNAseq to enrich the analysis with our cells of interest (Fig. 1a). To do so, TAT-CRE fusion protein was first stereotactically infused into the 3V of male tdTomato[loxP/+] Cre reporter mice. As TAT-CRE targets cells close to the infusion site[34], this injection allows for precisely labeling the 3V ependyma, including both typical

ependymal cells and tanycytes (Fig. 1a, Supplementary Fig. 1a), as previously described[15]. One week later, MBH microdissections were harvested at 8−9 a.m. from mice under different metabolic conditions (i.e., fed, 12h-, and 24h-fasting) (Fig. 1a, Supplementary Fig. 1b). After cell dissociation, TdTomato-positive cells were sorted by FACS with a loose gating strategy to optimize the collection of ependymocytes at the risk of sorting a few neighboring cells (Fig. 1a, Supplementary Fig. 1c). Our ependyma-enriched cell suspension was finally processed by Chromium (10x Genomics) to obtain the transcriptional profile of 13,121 individual cells (3261 in fed, 4944 in 12h-fasting, 4916 in 24h-fasting) (Fig. 1a, Supplementary Data 1a, b).

Unsupervised clustering analysis was first performed from the UMAP embedding and identified 21 distinct subpopulations (19 in fed and 21 in 12h- and 24h-fasting) (Fig. 1b, Supplementary Fig. 1d, e, Supplementary Data 1b). Following the identification of their enriched features and comparing them with known marker genes, we assigned a single identity to each cluster: tanycytes (three clusters; *Rax* + ), typical ependymal cells (two clusters; *Ccdc153* + , *Tmem212* + ), vascular and leptomeningeal cells (VLMC) (two clusters; *Dcn* + ), neurons (four clusters; *Tubb3* + ), blood vessels (two clusters; *Slco1c1*+ and *Exoc3l2* + ), pericytes (three clusters; *Art3* + ), microglia (one cluster; *Aif1* + ), pars tuberalis (two clusters; *Pitx2* + ), astrocytes (one cluster; *Aldh1l1* + ), oligodendrocytes (one cluster; *Olig1* + ) (Fig. 1c, Supplementary Data 2a). Regarding tanycyte subgroups, a further allocation may be done based on our current classification: "tanycytes1" express *Crym* and *Ephb1* defining β1/α2-tanycytes; "tanycytes2" express *Rspo3* and *Ntsr2* defining α1-tanycytes; and "tanycytes3" express *Fndc3c1* and *Col25a1* defining β2-tanycytes[35,36] (Fig. 1c, Supplementary Data 2a).

This unsupervised clustering analysis first confirms a 41.8% enrichment in both typical ependymal cells and tanycytes in our single-cell suspension, the other populations representing less than 11% per cell type (Supplementary Data 1b). Moreover, the analysis of tdTomato expression in our dataset shows that TAT-Cre injection into the 3V mainly targets the ependyma, consistent with our previous reports[15,18]: 78.3% of tdTomato-expressing cells are ependymocytes, whereas other populations represent less than 3% per cell type (Supplementary Fig. 1e, Supplementary Data 1c). The presence of other cell types likely arises from the gating strategy and/or cross-contamination by tdTomato-positive cell debris during tissue dissociation. Indeed, tanycytes make numerous contacts with various cell types in the MBH[15], which may be kept by our soft dissociation, leading to their isolation. Finally, the hierarchical structure of the clusters reveals the impact of spatial organization and microenvironment on the cellular transcriptional profile, which appears to be organized into four categories: the parenchymal cells (e.g., neurons and microglia), the pars tuberalis, the vascular system (e.g., pericytes, endothelial cells, and mural cells), and the ependyma (e.g., tanycytes and typical ependymal cells) (Fig. 1c).

### Ependymocytes form a heterogeneous population along the 3V
Our FACS-associated scRNAseq approach initially aimed to characterize transcriptomic profiles of the ependymocytes lining the MBH. We next subsetted the data to focus on tanycytes, typical ependymal cells, and astrocytes in the fed condition to reveal a finer heterogeneity potentially hidden in the initial clustering workflow (Fig. 1d, Supplementary Data 1d). After re-clustering using the same parameters, seven clusters were detected in the fed state, including two typical ependymal cell clusters, one astrocyte cluster, and four tanycyte clusters (Fig. 1d). First, typical ependymal cells are organized in two distinct populations, namely Epen1 (*Krt15* + /*Col6a5* + ) and Epen2 (*Barhl2* + /*Pitx2* + ) (Fig. 1e). Interestingly, some features, such as *Ccdc153* or *Tmem212*, are shared with similar expression levels between these two subgroups, whereas others display either different expression levels (e.g., *Ascc1* is enriched in Epen1, whereas *Csrp2* is in Epen2) or specificity (e.g., *Krt15* and *Barhl2* are specific to Epen1 and 2, respectively)

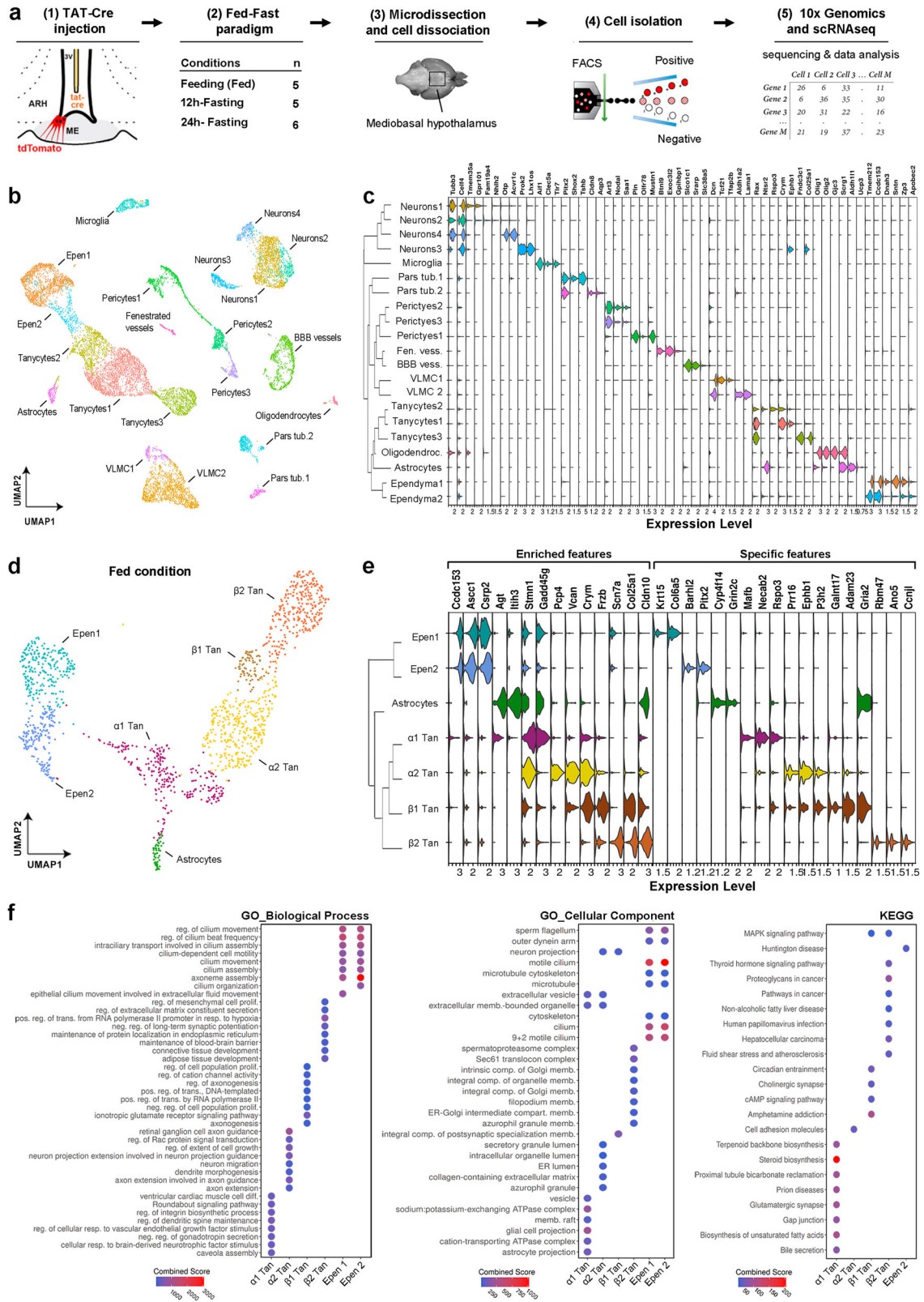

(Fig. 1e, Supplementary Data 2b). However, by cross-referencing each specific feature with in situ hybridization data from the Allen Mouse Brain Atlas (available from https://mouse.brain-map.org/)[37], no well-defined regions are found along the 3V for these two typical ependymal populations but rather a sparse and interlaced distribution (Supplementary Fig. 2a). Of note, these populations do not match the two ependymal populations previously identified by Campbell et al.[35].

Moreover, gene ontology (GO) enrichment analyses to further characterize their main biological processes, cellular components, and KEGG reveal no clear and distinct functions between these two populations (Fig. 1f, Supplementary Data 2c–g), suggesting that typical ependymal cells are relatively homogeneous.

Regarding the tanycyte population, known markers were first used to identify the four current subtypes, named β2-, β1-, α2-, and α1-

**Fig. 1 | High-resolution transcriptomic profiling of MBH ependymal cells.**
**a** Experimental workflow. TAT-Cre was infused into the 3V (1) to target the ependyma. One week later, mice were sacrificed in three metabolic conditions (n = 5 male mice per condition) (2). The mediobasal hypothalamus (MBH) was microdissected (3); cells were then dissociated, recovered by FACS (4), and processed by 10 × 3′ whole transcriptome scRNA sequencing (5). **b** UMAP representation of the three integrated conditions (i.e., fed, 12h-fasting, and 24h-fasting) for a total of 13,121 cells, colored per cluster and annotated according to known cell types.
**c** Dendrogram showing the hierarchical clustering between populations and violin plot displaying known markers and specific features per cell type. Specific features are based on the ratio between the percentage of expressing cells in the given population vs. all other cells. **d** Subset and re-clustering of ependymal populations

in the fed condition, colored per cluster and annotated according to known cell types. **e** Dendrogram showing relatedness between clusters and violin plot displaying enriched (left) vs. specific (right) features per cell type. Enriched features are based on differential gene expression analysis. Specific features are based on the ratio between the percentage of expressing cells in the given population vs. all other cells. **f** Cluster characterization with Gene Ontology (GO) highlighting molecular function, cellular component, and KEGG. The top 8 significant terms are displayed and classified by Combined Score. BBB blood-brain barrier, Epen typical ependymal cells, FACS Fluorescence Activated Cell Sorting, GO gene ontology, Tan tanycytes, VLMC vascular and leptomeningeal cells. See Supplementary Figs. 1, 2 and Data 1, 2. Images in panel (**a**) are adapted from ref. 18.

tanycytes (Fig. 1d, e). GO enrichment analyses confirmed most of the known tanycyte functions (Fig. 1f, Supplementary Data 2c–g). For instance, β2-tanycytes express genes related to the maintenance of the blood-brain barrier (Fig. 1f), primarily associated with vascular development, such as *Vegfa*, and tight or adherens junctions, such as *Cldn10, Cldn3, Jam2*, and *Tjp1* (Fig. 1e, Supplementary Data 2b)[18,20,38]. We also detected the expression of genes related to the thyroid hormone signaling pathway, such as *Dio2*[24,39], or MAPK/ERK signaling pathway[23,40]. Additional biological functions, such as extracellular matrix, endoplasmic reticulum, and Golgi compartments, are also highlighted. Notably, β2-tanycytes express the connective tissue growth factor (*Ctgf*), an extracellular matrix-associated heparin-binding protein binding different growth factors, such as VEGF[18] or TGF-β[41], known to be involved in ME plasticity (Supplementary Data 2b). Regarding the transcriptomic profile of β1-, α2-, and α1-tanycytes, many genes and GO terms are shared and appear to be involved in neuronal processes, especially axonal guidance, dendrite morphogenesis, and synaptic functions, highlighting tanycyte-neuron interactions (Fig. 1e, f, Supplementary Data 2c–g). Moreover, α1 tanycytes appear to be the closest population to astrocytes, expressing genes related to astrocyte projection (Fig. 1e, f).

Nonetheless, clustering analysis of ependymal populations revealed some critical limitations. First, the most enriched features -calculated by differential gene expression- do not well-define ependymal subgroups along the 3V (Fig. 1e, Supplementary Fig. 1h, Supplementary Data 2b), as confirmed by in situ hybridization data available on the Allen Mouse Brain Atlas[37] (Supplementary Fig. 2b, c). Indeed, while several markers well define typical ependymal cells or β2-tanycytes, which constitute "stable" subgroups, enriched markers for β1-, α2-, and α1-tanycytes span over several clusters (Fig. 1e). Similarly, specific features -calculated by the percentage ratio between expressing cells in the given population versus all others- fail to define β1-, α2-, and α1-tanycytes (Fig. 1e, Supplementary Data 2b). Conversely, we observed that many genes shape ventro→dorsal or dorso→ventral patterns along the 3V while overlapping different subgroups (Supplementary Fig. 1h), as validated by in situ hybridization data (Supplementary Fig. 2c). These shared markers partly explain the lack of functional specificity found in our GO analysis for the β1- and α-tanycyte populations (Supplementary Data 2c–g).

Altogether, although we can identify known ependymal subpopulations, clustering analysis fails to clearly define the boundaries of the different subpopulations, notably β1-, α2-, and α1-tanycytes, raising questions regarding our current ependymal classification[8].

## Supervised and unsupervised pseudospatial analyses dissect 3V complexity
To add an analytical and complementary perspective to the standard clustering analysis regarding the characterization of the tanycyte population, we devised a pseudospatial analysis (PSA) to explore the gradual pattern of gene expression along the 3V (Fig. 2a, b). Indeed, the neuroanatomical distribution of tanycytes is crucial in establishing their identity and functions, as they project in different brain nuclei

and contact various partners. Intriguingly, the UMAP distribution of ependymocytes mimics the anatomical ventrodorsal axis for the different ependymal subgroups (i.e., β2→β1→α2→α1→typical ependymal cells) (Fig. 1b, d) and follows the gradual patterns in gene expression of many known features, such as *Rax or Rarres*[42] (Supplementary Figs. 1g and 2). It is worth noting that similar UMAP distributions for 3V ependyma are also detected in other datasets, notably in the mouse HypoMap[43]. Taking advantage of this characteristic, we used Monocle3 to sort the cells in the UMAP space following what we interpreted as a pseudospatial (PS) trajectory from typical ependymal cells to β2 tanycytes (Fig. 2a). This analysis allowed us to reorganize and classify genes according to these PS trajectory (Fig. 2b).

To first define relevant markers for the current subpopulations, we divided the PS trajectory by distinguishing β2-, β1-, α2-, α1-tanycytes, and typical ependymal cells based on the UMAP clustering in the fed condition. Because the two typical ependymal populations share most of their features (i.e., up to 70%), they were considered a single population for the rest of the analysis. We then applied a masked correlation analysis to differentiate features highly expressed in one population (i.e., specific markers) from those spanning over multiple populations (i.e., shared markers) (Fig. 2c). Our approach identifies 631 specific features (e.g., *Pcp4l1, Ccdc153, Tctex1d4*) in typical ependymal cells and 129 in β2-tanycytes, confirming the robustness of these two subgroups. However, very few genes exclusively correlate to α1 (15 features: e.g., *Fcgr2b, Mafb, Eef1a2*), α2 (3 features: i.e., *Col6a3, Sema3a, Pcp4*), and β1 (0 feature) tanycytes along the PS trajectory (Fig. 2d, Supplementary Data 3a). Conversely, our analysis shows that most identified features correlate to multiple populations (Fig. 2d), constituting shared markers. Indeed, we identify 179 features overlapping between typical ependymal cells, α1-, and α2-tanycytes, 70 features between α1-, α2-, and β1-tanycytes, and 169 features between α2-, β1-, and β2-tanycytes (Fig. 2d). Strikingly, most features initially revealed by the standard clustering analysis account for shared markers with the supervised PSA (Supplementary Fig. 3a, Supplementary Data 3b). Next, gradual patterns in gene expression were validated using in situ hybridization: the distribution of known (e.g., *Cdh2, Col25a1, Trhr*) and newly identified (e.g., *Sult2, Nrxn1, Sema3f, Npr1, Phldb2*) features demonstrates the relevance of our PS analysis (Fig. 2e, f, Supplementary Fig. 3b, c). To further validate our analytic approach, the supervised PSA workflow was applied to a subset of the HypoMap dataset[43] containing only ependymal cells and tanycytes from the fed condition: it yields similar outcomes, notably the importance of shared features for α1-, α2-, and β1-tanycytes (Supplementary Fig. 3d–g, Supplementary Data 3g, h). Finally, the MERFISH dataset (obtained from the Allen brain atlas available at https://portal.brain-map.org/atlases-and-data/bkp/abc-atlas) was analyzed to study gene expression along the 3V: similar trajectories are obtained between the PSA and spatial transcriptomics, notably for *Serpine2* and *Gli3* (Supplementary Fig. 4a, b).

Based on our feature classification (specific vs. shared markers), we next performed GO enrichment analysis, notably for β2 tanycytes' specific markers and for α1-, α2-, and β1-tanycytes' shared markers. While known functions, such as the VEGF signaling pathway as the top

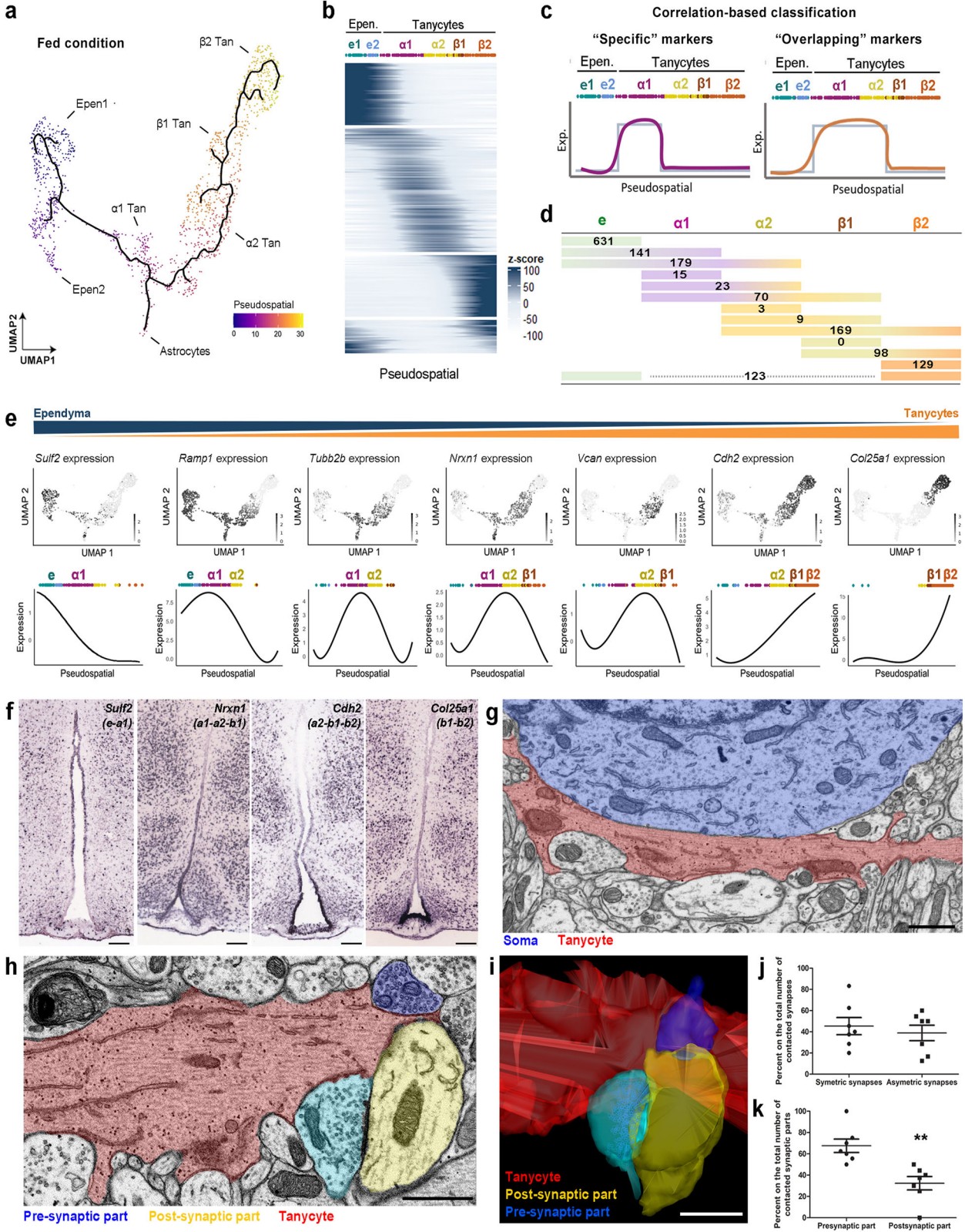

KEGG term[18], are found for β2-tanycyte, multiple terms related to lipid metabolism are also highlighted, notably postprandial triglyceride response (GWAS), waist circumference adjusted for BMI in smokers (GWAS), cholesterol metabolism (KEGG), or regulation of lipolysis in adipocytes (KEGG) (Supplementary Data 3c). Regarding GO analysis of shared features among α1-, α2-, and β1-tanycytes, most of the common functions include neural developmental processes and synaptic

organization, notably the post-synaptic compartment (Supplementary Data 3c), highlighting once again tanycyte-neuron interactions. To validate such interactions, we next examined tanycyte processes facing the ARH-VMH using electron microscopy, as done in our previous studies[15]. 3D-reconstructions obtained from a series of hypothalamic sections allowed us to visualize numerous tanycyte-soma (Fig. 2g) and tanycyte-synapse contacts (Fig. 2h, i): the interactions

**Fig. 2 | Fine heterogeneity of the 3V ependyma revealed by supervised pseudospatial analysis. a** UMAP representation displaying the pseudospatial trajectory from typical ependymal cells to β2-tanycytes. **b** Heatmap showing gradual gene expression in the typical ependymal cells→β2-tanycytes trajectory. **c** Summary of the analytic workflow. Correlated-based classification allows the identification of features specific to one subgroup vs. shared by two or three subgroups. **d** Number of features significantly correlating with one vs. multiple ependymal populations. **e** Feature plots showing the normalized expression and pseudospatial expression of some shared genes following the typical ependymal cells→β2-tanycytes trajectory. **f** In situ hybridization on coronal brain sections from Allen Mouse Brain Atlas validating the gradual gene expression distribution along the 3V (scale bars = 150 μm).

Inverse contrast scanning electron microscopy micrograph showing a tanycyte (red)/soma (blue) interaction (**g**) and a tanycyte (red)/presynaptic part (yellow)/ post-synaptic part (blue) interaction (**h**) (scale bars = 1 μm). **i** 3D representation of the tanycyte/synapse interaction shown in (**h**) (scale bar = 1 μm). **j** Percentage of tanycyte interactions with symmetric vs. asymmetric synapses. A total of 7 tanycytes in 4 different fed male mice were analyzed (two-tailed t-test; $p = 0.5650$). **k** Percentage of tanycyte interactions with pre-synapse vs. post-synapse. A total of 7 tanycytes in 4 different fed male mice were analyzed (two-tailed t-test; $p = 0.019$). In (**j**, **k**), data are means ± SEM. **$p < 0.01$ compared to "presynaptic part". Epen (or **e**), typical ependymal cells; Tan, tanycytes. See Supplementary Figs. 2–4 and Data 3. Source data are provided as a Source Data file.

occur along the tanycyte process with both symmetric (i.e., GABAergic) or asymmetric (i.e., glutamatergic) synapses (Fig. 2j). However, the presynaptic terminals appear to be the preferred contact sites (Fig. 2k), consistent with our GO analysis (Supplementary Data 3c).

To further challenge our current classification and identify boundaries along the 3V only based on gene expression, we next performed an unsupervised PSA using TradeSeq[44] (Fig. 3a). This analysis allowed us to group sets of genes that follow similar expression patterns along the PS trajectory (Fig. 3a, Supplementary Data 4a). In the fed condition, twenty-three patterns containing more than 50 genes were identified (Fig. 3b, Supplementary Fig. 4c). For each pattern, we determined "transitional zones" as critical points within the PS trajectory where the gene expression reaches 0 (Fig. 3a). Finally, we combined these different transitional zones to define "split regions" along the 3V (Fig. 3a). This approach −only based on an unsupervised analysis of gene expression gradients− finds two main split regions along the 3V, representing a division in three ependymoglial subgroups (Fig. 3c). These subgroups define a β-tanycyte population, a mix of α1- and α2-tanycytes, and an ependymal population (Fig. 3c), matching our previous supervised PSA which revealed fuzzy boundaries between α1-, α2-, and β1-tanycyte subpopulations. Therefore, these populations must preferably be considered a "continuum parenchymal tanycyte" group rather than distinct subgroups (Fig. 3d).

Finally, both supervised (Fig. 2) and unsupervised (Fig. 3) PSA also reveal that ventrodorsal expression gradients are not the only patterns along the 3V: indeed, some features can be shared between non-neighboring populations, drawing U-shape patterns along the PS trajectory (Fig. 3e, f). Notably, we identified 123 features concomitantly correlating to ependymal cells and β2-tanycytes (Fig. 2d, Supplementary Data 3a) and validated some of them using the Allen Brain Atlas (Fig. 3g).

## Fasting increases gene expression dynamics along the 3V

In addition to their heterogeneity, tanycytes are highly plastic cell types and display changes in gene expression during energy imbalance[18,27]. To identify dynamics in gene expression profile during the fed→fasting paradigm, we applied our three complementary analytic approaches, namely standard clustering analysis, supervised PSA, and unsupervised PSA, on the 12h-fasting and 24h-fasting datasets focusing on the ependyma with the same parameters used for the fed condition (Supplementary Fig. 5, Supplementary Data 2−4).

Concerning the standard clustering analysis, tanycyte subgroups increase heterogeneity as we advance in the fasting paradigm. Indeed, five tanycyte clusters are defined in the 12h-fasting state and six in the 24h-fasting state (Supplementary Fig. 5a−c). Similarly, four typical ependymal cell clusters are found in the 24h-fasting versus only two in the fed and 12h-fasting states (Supplementary Fig. 5a). In particular, the α1-tanycyte cluster splits in two during the fed→12h-fasting transition, whereas the α2 population divides in two during the 12h-fasting→24h-fasting transition (Supplementary Fig. 5b). The expression of numerous ribosomal genes (e.g., *Eef1a2, Rpl3, Rpl18, Rpl27a*) increases in 12h- and 24h-fasting in the α1.2 and α2.2 populations (Supplementary Data 2h), respectively, suggesting a high degree of translation at those

metabolic stages in a subset of cells. Additionally, the analysis of the common features shows that α2.1-tanycytes in the 24h-fasting condition share more features (e.g., *Emd, Btg2,* or Nr4a1) with β1- and β2-tanycytes than during the fed state (Supplementary Data 2h, i), suggesting a dynamic shift of β-tanycyte phenotype towards α-tanycytes. Of note, many of these shared features are TFs belonging to the AP1 transcriptional complex. To summarize, α1- and α2-tanycytes, which were previously identified with β1 as the continuum parenchymal tanycyte group, display the most heterogenous and plastic phenotype during the fed→fasting transition (Supplementary Fig. 5h, i).

The supervised PSA provides a deeper view regarding the shift of markers along the PS trajectory in the different metabolic conditions. Indeed, specific features decrease while shared features increase at 24h-fasting (Supplementary Fig. 5d, e, Supplementary Data 3a), confirming dynamic shifts between subgroups. Strikingly, supervised PSA reveals the loss of specificity for β2-tanycytes: no specific markers are found at 24h-fasting anymore. In contrast, numerous specific markers in the fed condition (around 20%), such as *Adm, Cacna2d2, Fndc5,* and *Scn7a*, become β1-β2 shared features at 24h-fasting, suggesting an extension of β2 phenotype to the β1 subgroup (Supplementary Data 3a).

To further analyze such shifts in gene expression gradients without the constraints of predefined boundaries along the 3V, we finally applied the unsupervised PSA (i.e., sets of genes following similar expression patterns along the PS trajectory) (Supplementary Fig. 5f, g, Supplementary Data 4a). In agreement with the previous analyses, an increase in the number of gene expression patterns is observed in 12h-fasting and 24h-fasting conditions (i.e., from 23 patterns in fed to 32 and 34 patterns in 12h-fasting and 24h-fasting conditions, respectively) (Supplementary Fig. 5f, Supplementary Data 4a), suggesting once more an increased gene expression dynamics during fasting. When analyzing the split regions along the 3V based on pseudospatial-ordered gene expression, only one critical point emerges at the level of α1-tanycytes at 24h-fasting (Supplementary Fig. 5g), showing that high gene expression dynamics and shifts along the 3V at this stage blurs the boundaries of the classical tanycyte classification (Supplementary Fig. 5h, i).

## Functional protein → RNA processing switch during fasting

Given the gene expression dynamics induced by the metabolic states, we first applied a pseudobulk differential gene expression analysis (DGEA) on defined clusters to study the changes in gene expression profile in tanycytes between fed and fasting conditions. In the integrated dataset (i.e., including all metabolic conditions together) (Fig. 4a), we identified hundreds of genes significantly up- or down-regulated in response to fasting (Fig. 4b, Supplementary Data 5a−d). Surprisingly, very few transcriptional differences are observed in the astrocyte cluster, suggesting those cells are less affected by metabolic changes, whereas α2-tanycytes appear to be the more dynamic population (Fig. 4b). Within this population, transcriptional changes in genes related to RNA synthesis and processing are strikingly upregulated during fasting, notably for immediate-early TFs (e.g., *Egr1, Egr2, Fosb, Nr4a1, Nr4a3, or Stat3*) and RNA binding proteins present in the

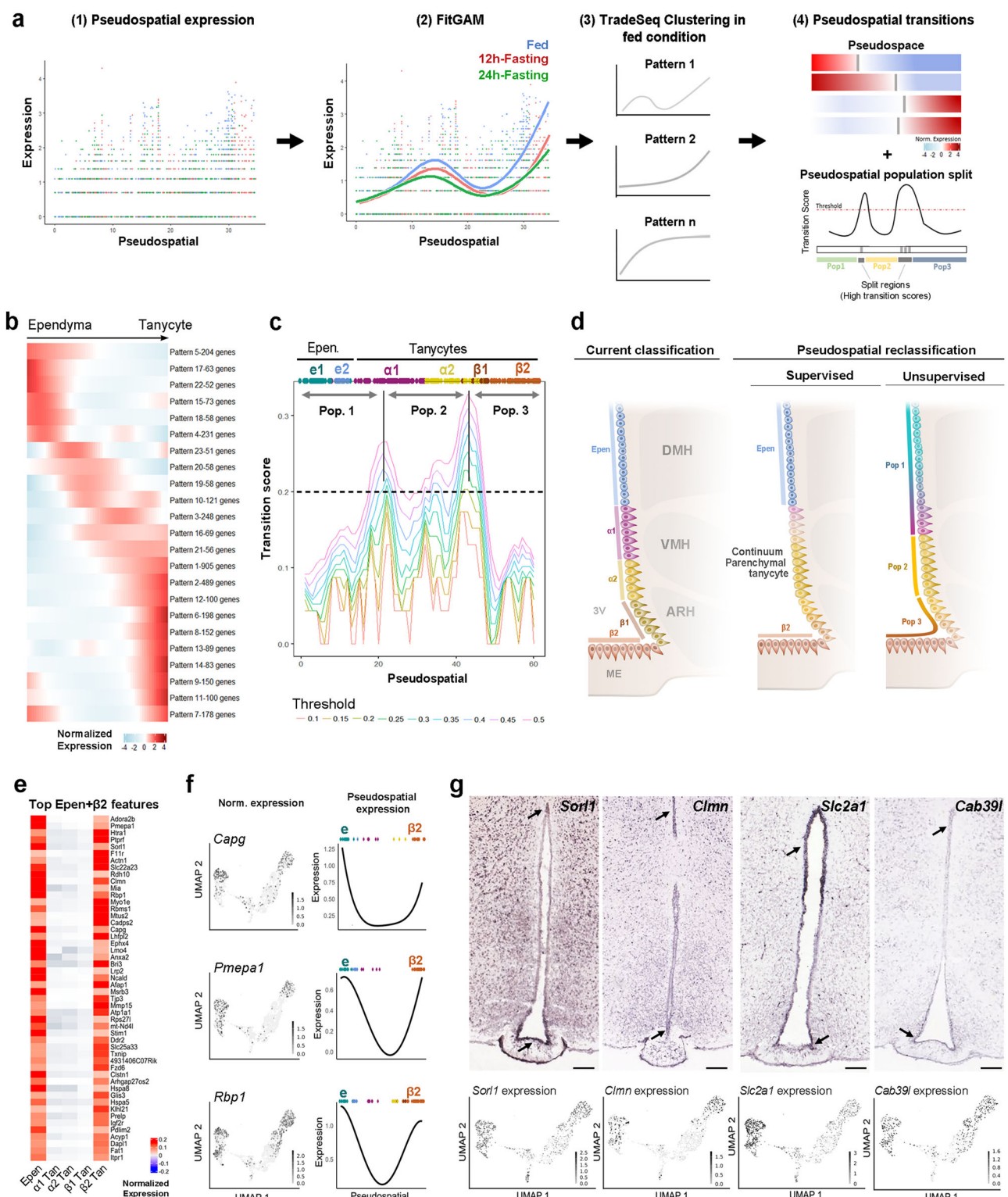

nucleus and the spliceosome (e.g., *Luc7l3*, *Rbm3*, *Srsf2*, *Srrm2*) (Fig. 4c, Supplementary Data 5b–d). Additionally, genes involved in the thyroid hormone signaling pathway and synthesis (e.g., *Dio2*, *Rcan1*, *Notch2*) are also upregulated during fasting in the α2-tanycytes (Fig. 4c, Supplementary Data 5b–d), consistent with previous reports[27,39]. Regarding the other tanycyte populations (Supplementary Data 5b–d), *Cldn10* mRNA increases in β2- and β1-tanycytes in 12h-fasting, consistent with the rise in tanycyte barrier tightness observed during fasting[18].

While many genes are differentially regulated in specific subtypes, several genes change similarly across the different tanycyte

populations. For instance, during 12h-fasting, 131 genes are significantly upregulated (FDR < 0.05) in both β2- and α2-tanycyte subgroups (Supplementary Data 5e). Notably, aldolase (*Aldoc*), an enzyme involved in glycolysis/gluconeogenesis, is the most upregulated gene in all cell types in the 24h-fasting state (Fig. 4d), suggesting changes in metabolic pathways. Similarly, *Mt1* and *Mt2* genes, encoding metallothionein-1 and 2, are upregulated in all populations after 24h-fasting (Supplementary Data 5e). These low molecular weight proteins rich in cysteine bind divalent heavy metal ions and play several putative roles in metal detoxification, Zn and Cu homeostasis, and

**Fig. 3 | 3V segmentation by unsupervised pseudospatial analysis. a** Summary of the analytic workflow. Pseudospatial gene expression profiles (1-2) were unsupervisedly organized and regrouped into patterns using TradeSeq (3). The pseudospatial transitions for each pattern (gray bars in (4)) were added to determine key "split" regions (4). **b** Heatmap showing the twenty-three pseudospatial gene expression patterns (containing more than 50 genes) along the top→bottom trajectory. **c** Graph representing the main transition region along the pseudospatial trajectory. The transition score was calculated as an on-off switch in gene expression along the trajectory. Different curves represent the different on-off gating thresholds used for the analysis. **d** Improved model for tanycyte classification based on the supervised and unsupervised pseudospatial analysis. Supervised pseudospatial analysis describes β2 tanycytes and typical ependymal cells as stable populations and a continuum parenchymal tanycyte group with gradual changes in gene expression. Unsupervised pseudospatial analysis also describes three main populations along the 3V in the fed condition. Heatmap (**e**) and feature plots (**f**) showing normalized expression and pseudospatial (PS) expression of representative genes following the U-shape pattern (i.e., expression in typical ependymal cells and β2-tanycytes). Within the genes exhibiting this PS distribution, we notably found *Capg*, *Pmepa1*, and *Rbp1*, genes involved in growth factor binding, lipid binding, and endosome membrane. **g** In situ hybridization on coronal brain sections (Allen Mouse Brain Atlas) and feature plots derived from UMAP validating the gene expression and distribution of different features (Scale bars = 150 μm). Features were selected for their U-shape pattern along the 3V. See Supplementary Fig. 4 and Data 4.

scavenging free radicals. Interestingly, disrupting these two metallothionein genes in mice resulted in obesity[45]. Alternatively, we also observe a few genes oppositely regulated in different subpopulations. For instance, *Nrxn1* is downregulated in α tanycytes but upregulated in β tanycytes during fasting (Fig. 4d, Supplementary Data 5e), suggesting a dorso→ventral shift in gene expression along the 3V.

During energy imbalance, the temporal factor plays a crucial role: different biological processes may be activated at 12h-fasting versus 24h-fasting. To study this, we next analyzed gene expression dynamics in tanycyte subpopulations through the fed→fasting temporal trajectory and finally performed GO analyses (Fig. 4e, f, Supplementary Data 5e). For the α2-tanycytes, genes related to asymmetric synapse and autolysosome increase with fasting time course (i.e., I1 + I4 or I1 + I2 + I3 + I4 combinations). Conversely, endoplasmic reticulum function, protein processing, and vesicular functions are downregulated with fasting time course (i.e., D1 + D4 or D1 + D2 + D3 + D4 combination), suggesting that RNA processing replaces protein processing (Fig. 4f, Supplementary Data 5f–j). Interestingly, the transitional temporal trajectories (i.e., I6 + T6 + D6 combination for an upregulation and I5 + T5 + D5 combination for a downregulation at 12h-fasting, respectively) indicate a stepwise functional reorganization: genes involved in mRNA processing, especially RNA splicing, are upregulated only at 12h-fasting, while those engaged in ribosomal assembly, RNA stability and translation increase at 24h-fasting (Fig. 4f).

### Fasting redistributes gene expression gradients along the 3V

To study the transcriptional shifts in gene expression along the 3V induced by the fed→fasting paradigm, we next performed a differential pseudospatial gradient analysis (DPSA) on the integrated dataset (Fig. 5). As for each isolated metabolic condition, the integrated PS trajectory follows the neuroanatomy from typical ependymal cells to β2 tanycytes, and cells from different feeding and fasting conditions appear uniformly intermingled in the integrated UMAP representation, indicating that the transcriptional identities of these trajectories are stable across those experimental conditions (Fig. 5a). To compare the metabolic conditions, we analyzed the distribution of the 23 fed gene expression patterns in the 12h- and 24h-fasting conditions (Fig. 5b). While the fed and 12h-fasting conditions display similar gene expression distribution along the 3V, 24h-fasting induces changes in numerous patterns, including both ventro→dorsal (e.g., pattern 13) and dorso→ventral shifts (e.g., patterns 19, 20, and 23) along the 3V (Fig. 5b). To validate such shifts, we performed in situ hybridization using RNAscope in fed versus 24h-fasting conditions for some genes. Regarding *Deptor* (pattern 1), a gene involved in mTOR signaling, the DPSA reveals a dorso→ventral shift during fasting (Fig. 5c): similar results are obtained using in situ hybridization, showing that the gene is restricted to the β-tanycytes during fasting (Fig. 5d, e, Supplementary Fig. 6a).

Given the significant upregulation of TFs observed during fasting (i.e., *Egr1*, *Egr2*, *Fosb*, *Nr4a1*, *Nr4a3*, or *Stat3*) (Fig. 4c), with notable shifts from β- to α2-tanycytes (Supplementary Data 2h, i),

transcriptional activity may be responsible for this transcriptional rearrangement, resulting in shifts in tanycyte gene profile observed along the 3V. To further investigate the activity of TFs in the ependyma, we performed a SCENIC analysis[46] in the three metabolic conditions on the PS trajectory. First, the number of active regulons detected by the algorithm remarkably increases from fed to fasting (85 in fed vs. 424 in 12h-fasting vs. 427 in 24h-fasting), highlighting once again a dynamic change of gene expression in these conditions. Second, the activity of the 59 regulons shared among all three states −represented as a PS-ordered heatmap (Fig. 5f)− defines three main distinct active regions along the 3V in the fed condition (Fig. 5g), confirming our reclassification into the ependymal cells, a continuum parenchymal tanycyte group encompassing α1, α2, and β1, and the β2 population (Fig. 3d). Strikingly, at 12h-fasting, many TFs, including the above-mentioned *Egr1*, *Egr2*, and *Fosb*, change their activity region from β2 to β1-α2-α1 while the ependymal specific transcriptional block remains stable (Fig. 5f, g). At 24h-fasting, this phenomenon is even more evident: the number of TFs presenting with a clear ependymal and β2 expression profile diminishes, while most TFs acquire a broader activation profile across all the subpopulations (Fig. 5f, g). This analysis confirms the transcriptional rearrangement induced by metabolic changes, unveiling the flexibility and adaptability of the parenchymal tanycyte population and the blurring of tanycyte subgroup boundaries during fasting.

### Spatial distribution of tanycyte phenotype and functions along the 3V

As tanycytes display high transcriptional dynamics in response to the metabolic state by changing and/or shifting their gene expression, we finally analyzed the consequences on tanycyte functions.

Focusing on patterns 19, 20, and 23, which exhibit 167 features with a similar dorso→ventral shift along the 3V during fasting, we first performed a GO enrichment analysis to identify potential functions that could move from α- to β-tanycytes. This analysis mainly reveals changes in cholesterol and lipid metabolism (Fig. 6a). Using immunohistochemistry, we first validated the dorso→ventral shift in FASN (pattern 19) expression, a protein involved in lipid biosynthesis (Fig. 6b): while FASN staining is present in α-tanycyte lining the VMH in the fed condition, it extends to the adjacent β1 tanycytes lining the vmARH during fasting (Fig. 6c, d, Supplementary Fig. 6b), confirming a phenotypic change for this subgroup. To functionally validate the impact of such shifts in gene expression on lipid metabolism, we also performed filipin staining, which binds free and unesterified cholesterol. In the fed condition, filipin staining is present along the 3V in tanycyte cell bodies lining the DMH, VMH, and dmARH (Fig. 6e, f), whereas it shuts down in the ependyma lining the DMH and extends to the vmARH during fasting (Fig. 6e, f, Supplementary Fig. 6d), confirming the dorso→ventral functional shift for cholesterol metabolism revealed by the DPSA (Fig. 6g).

Another tanycyte hallmark revealed in this study by the clustering analysis and the PSA, further validated by EM, was their interactions with neurons and synapses (Figs. 1–2). Here, the DPSA demonstrates

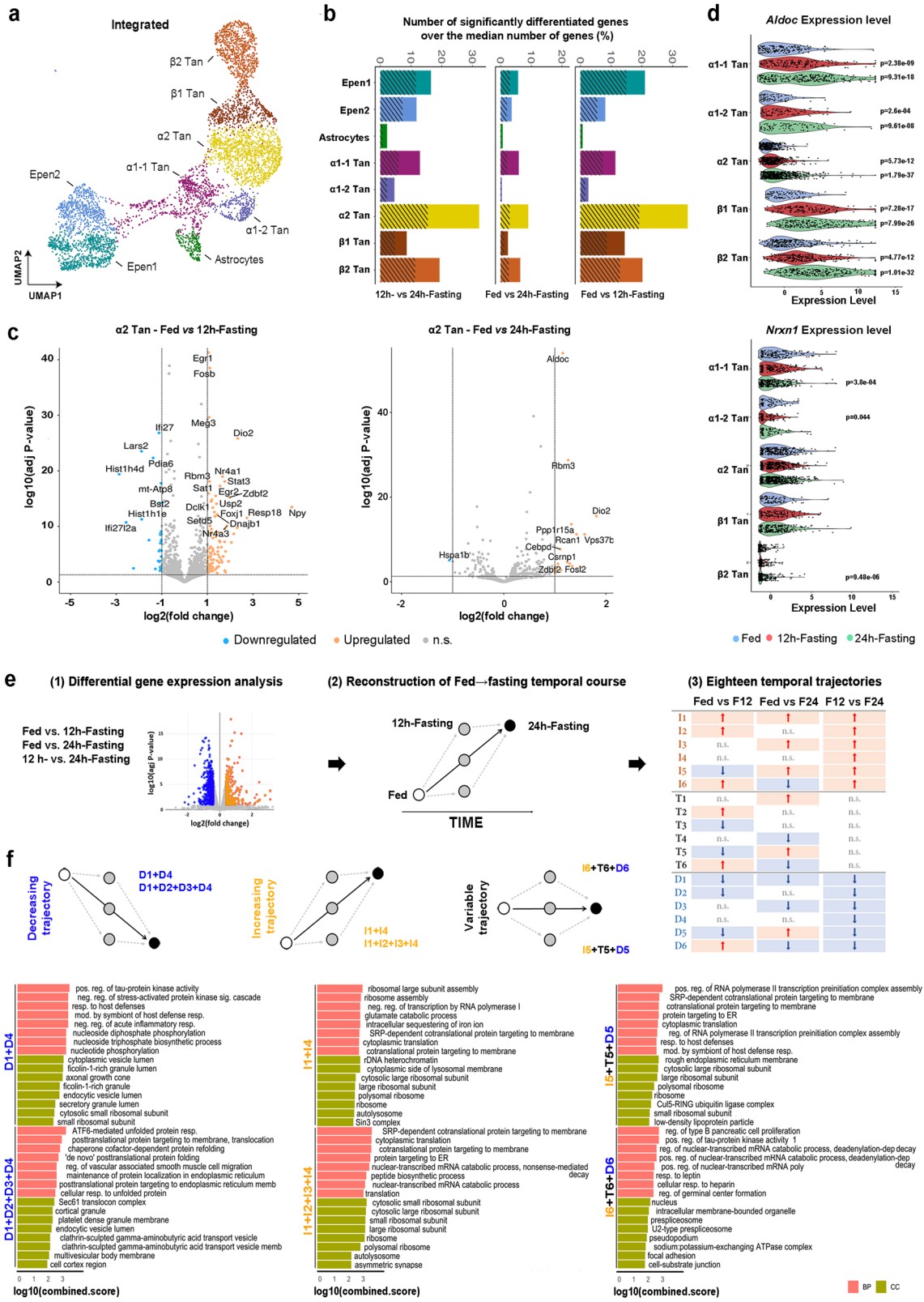

that this characteristic also presents a dorso→ventral shift during fasting (cluster 10) (Supplementary Data 4b). Such tanycyte-neuron interactions are likely crucial for controlling energy balance[17,28] and may explain tanycyte plasticity observed during fasting. Indeed, cells sense the presence of potential interaction partners through a wide range of receptors and, specifically, respond by changing the expression of many target genes via complex regulatory networks. To study

tanycyte-neuron interactions and their dynamics on the ventrodorsal axis during energy imbalance, we first developed a cell-cell communication analysis using a communication score matrix based on ligand-receptor expression products between tanycytes and neurons in the fed versus fasting conditions (Fig. 7a). To perform this analysis on the ventrodorsal axis, we used the classification from our initial UMAP (Fig. 1b), in which "tanycytes2" constitute the dorsal population

**Fig. 4 | Fed→Fasting temporal trajectories based on differential gene expression analysis reveal a cell state switch during energy imbalance. a** Integrated UMAP representation holding the three metabolic conditions together, colored per cluster and annotated according to known cell types. **b** Bar height representing the normalized proportion of differentiated genes calculated for each population: the number of differentially expressed genes (positive and negative) was divided by the median number of genes for each population. The shaded area of the bar represents the proportion of down-regulated genes. **c** Volcano plot displaying genes differentially expressed in 12h-fasting vs. fed and 24h-fasting vs. fed for the α2-tanycytes. **d** Violin plot of *Aldoc* and *Nrxn1* expression levels across multiple ependyma subpopulations in different metabolic conditions. Wilcoxon test was used to compare normalized expression levels in the fed condition against 12h-

fasting and 24h-fasting, respectively. **e** Summary of the analytic workflow. Differentially expressed genes for each comparison (1) were integrated to draw fed→fasting temporal course (2) and defined eighteen trajectories (3). **f** Trajectory characterization with Gene Ontology (GO) highlighting molecular function and cellular component in α2 tanycytes. The top 8 terms are displayed and classified by Combined Score. I1 + I4 or D1 + D4 trajectories were combined to reflect a continuous increase or decrease in gene expression, respectively. I1 + I2 + I3 + I4 or D1 + D2 + D3 + D4 trajectories were combined to reflect a global increase or decrease over time, respectively. I6 + T6 + D6 or I5 + T5 + D5 trajectories were combined to reflect an increase or decrease at 12h-fasting, respectively. See Supplementary Fig. 5 and Data 5.

(i.e., DMH/VMH tanycytes), "tanycytes1" the ventral population (i.e., ARH tanycytes), and "tanycytes3" the ME population (Fig. 7b, c). First, 24h-fasting mainly increases ligand-receptor communication related to extracellular matrix (ECM) signaling, whereas it downregulates those related to secreted signaling for all tanycyte subgroups (Fig. 7c, Supplementary Data 6a). Among the ligand-receptor couples involved in secreted signaling modulated by 24h-fasting, tanycytic pleiotrophin (*Ptn*) appears crucial. Pleiotrophin is a secreted heparin-binding growth factor involved in cell growth and survival, cell migration, and angiogenesis. Different functions may occur through its different receptors. In particular, *Sdc4* plays a role in adiposity and metabolic complications[47]. During fasting, ligand-receptor couples involving *Ptn* largely decrease in both tanycytes2→neurons and tanycytes1→neurons (Fig. 7b). Besides these changes shared by all tanycyte subgroups, regulation in intercellular communication may also differ from one tanycyte subgroup to another. Indeed, the downregulation in ligand-receptor couples related to cell-cell contacts is larger in tanycytes2 population (i.e., at the top of the ventricle) compared to tanycytes1 and tanycytes3 population (i.e., at the bottom of the ventricle) (Fig. 7c). Among ligand-receptor couples essential for cell-cell contacts, neurexin1 (NRXN1) and its receptors constitute some examples that are differently regulated between tanycyte populations (Fig. 7d–g). Belonging to shared features in α1-, α2-, and β1-tanycytes (Supplementary Data 3), neurexins are cell-surface receptors that bind neuroligins at both inhibitory and excitatory synapses in the central nervous system. *Nrxn1* is notably expressed by astrocytes in tripartite synapses[48,49]. In our cell-cell communication analysis, changes in the *Nrxn1*-receptor couples during fasting differ between tanycytes2 and tanycytes1 populations, demonstrating a shift in tanycyte-neuron communication (Fig. 7b, d). Indeed, *Nrxn1-Nlgn1* and *Nrxn1-Nlgn3* are downregulated in tanycytes2→neuron while *Nrxn1-Nlgn1*, *Nrxn1-Nlgn2*, and *Nrxn1-Nlgn3* are upregulated in tanycytes1→neuron interactions (Fig. 7b, d). These differential changes are consistent with the 3V dorso→ventral shift in *Nrxn1* expression found during fasting using our DGEA (Supplementary Data 5e) and DPSA (Supplementary Data 4a, pattern 10) (Fig. 7e). To validate such a shift in *Nrxn1* expression and inferred tanycyte-neuron communication, we further performed in situ hybridization: while *Nrxn1* mRNA is present in α-tanycyte lining the VMH and dmARH in the fed condition, its staining concentrates in β1 tanycytes lining the ARH during fasting (Fig. 7f, g, Supplementary Fig. 6c). Finally, to functionally validate the impact of such shifts in gene expression related to cell-cell contacts, we analyzed tanycyte-neuron and tanycyte-synapse interactions by immunohistochemistry (Fig. 7h, i), as previously described[15]. Our data indicates tanycytes projecting to the VMH/DMH lose numerous neuronal contacts with soma during fasting, whereas these contacts are maintained in the ARH (Fig. 7j). Additionally, we observe that tanycyte contacts with glutamatergic and GABAergic synapses also increase in the ARH (Fig. 7k, l). Altogether, these results show that the shift in gene expression that we characterized using our DPSA may have a critical impact on tanycyte→neuron interactions.

## Discussion

By using FACS-associated scRNAseq, this study deciphers the heterogeneity and plasticity of the 3V to better understand its functions in regulating energy balance. Focusing on adult mice, when the 3V is well-differentiated, we proposed a combination of analytic approaches, namely the standard clustering analysis and the pseudospatial trajectory analyses, to characterize cell populations and feature distribution along the ventricle. In this way, we first highlighted specific markers differentiating peculiar ependymal cell subgroups versus shared features overlapping in different populations. Taking the reasoning further, we delimited dynamic regions along the 3V and highlighted ventro→dorsal and dorso→ventral changes in PS gene expression patterns during a fed→fasting time course, showing that ependymal cells form a continuum that dynamically relocalizes its functions according to the metabolic status.

Within the MBH, the main characteristic of the 3V ependyma is its high heterogeneity, conferring its multifunctional biological properties. In this study, we aimed to reach a higher resolution of the MBH 3V in adult mice to further detail tanycyte classification with respect to previous analyses[35,36,43,50]. To this purpose, we adapted FACS-associated with scRNAseq, primarily used to study the developmental brain[51,52], to sort and focus on tdTomato-positive ependymal cells in adult mice in different metabolic conditions. Using a standard clustering analysis in the fed state, we detected the four historical tanycyte subgroups using previously described markers, allowing us to obtain a comprehensive marker list for each and highlighting known tanycyte functions. However, many markers appeared to display expression patterns exceeding the boundaries of their subpopulations, overlapping many subgroups. In particular, we highlighted that previously acknowledged markers of specific tanycyte subpopulations[35,36] are not limited to such clusters. Altogether, these weaknesses highlight the transcriptional complexity of ependymal cells and demonstrate the oversimplification of the current tanycyte classification[7,8]. To accurately explore tanycyte gene expression as gradients along the 3V axis, we used the distribution of ependymal cells in our UMAP dataset −which mimics the neuroanatomical distribution along the ventrodorsal axis from β2 tanycytes to typical ependymal cells− and developed a pseudospatial analysis (PSA). Defined as "pseudo" as the UMAP plot does not represent exact cell location in the intact tissue, this complementary analysis allowed us to explore the distribution patterns of known and undiscovered features without the constraints of predefined subgroups and to define alternative boundary areas along the 3V. Notably, our analysis revealed ventrodorsal and unusual top-bottom patterns, which likely play a crucial role in microdomain functions: in contrast, no clear anteroposterior patterns were identified despite previous reports[8,27,53], likely reflecting the limits of our UMAP pseudo-spatial representation along this dimension. Thus, PSA provides an added value to characterize cells from complex tissues while integrating spatial information and urges us to consider the ependyma as a continuum cell population, especially regarding the current α1-, α2-, and β1-tanycytes.

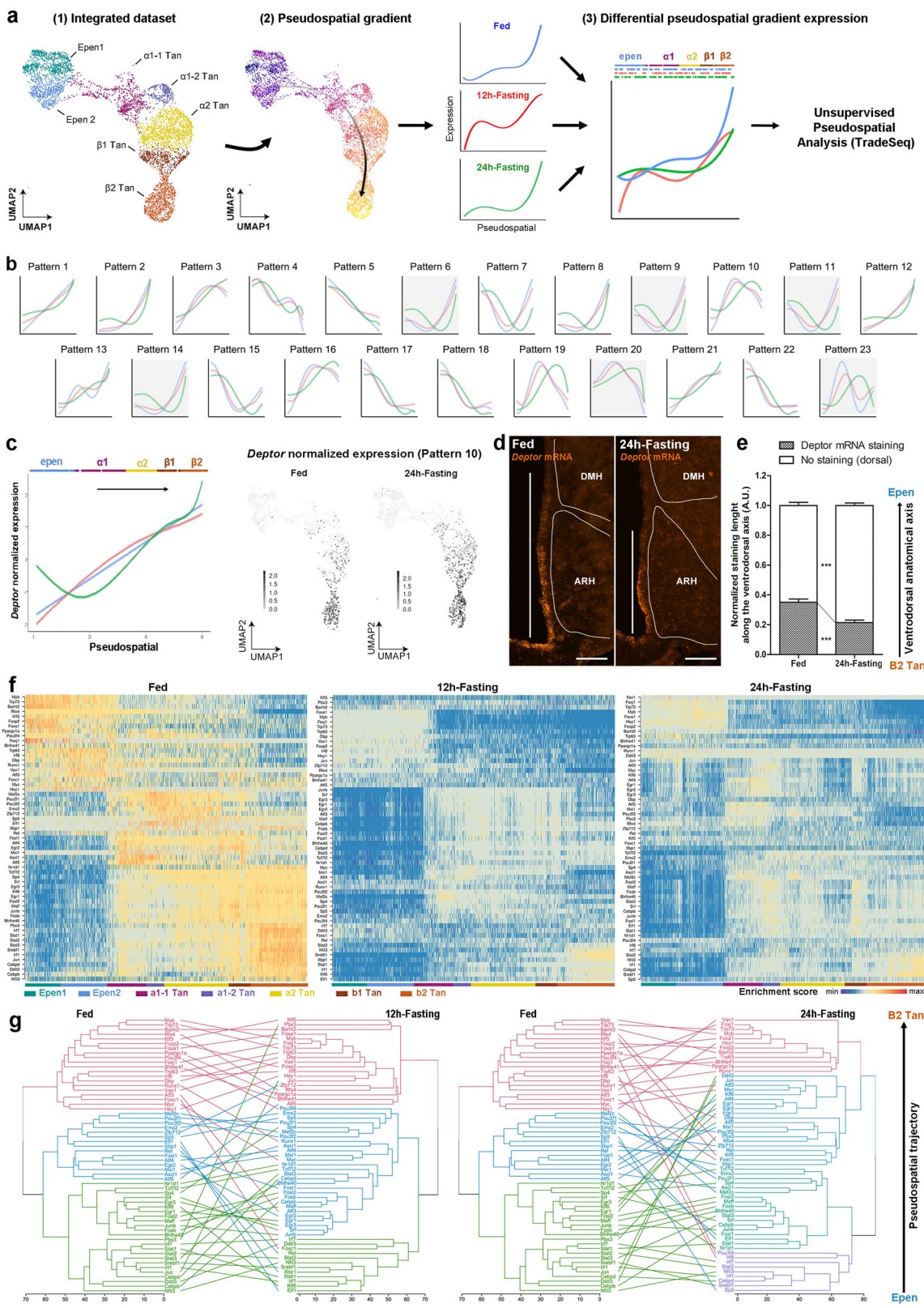

Combining standard clustering and pseudospatial analyses allowed us to further characterize the tanycyte functions. First, while our results did not suggest tanycyte-derived neurogenesis in the adult brain, as they do during postnatal ages[29,30], genetic manipulations[51] or neural injury[54], a peculiar trajectory toward astrocytes was observed in our dataset, as described by others[51], suggesting potential astrogenesis. Additionally, the continuum parenchymal tanycytes (i.e., α1-,

α2- and β1-populations) expressed numerous genes related to synaptic structure and functions and belonging to ligand-receptor couples for tanycyte-neuron communication. This result is supported by our recent data showing that tanycytes contact neurons and synapses in the MBH and receive synaptoid contacts along their processes[15]. Such molecular contacts also suggest that tanycytes function as astrocytes in tripartite synapses and can modulate neuronal firing activity as

**Fig. 5 | Differential pseudospatial analysis reveals gene expression shifts between ependymal populations during energy imbalance. a** Summary of the analytic workflow. After defining the pseudospatial trajectory from typical ependymal cells to β2-tanycytes in the integrated dataset (1), the difference in the pseudospatial gradient expression between conditions was calculated by TradeSeq (2). **b** Main patterns in gene expression along the pseudospatial trajectory defined in the fed condition (blue) and compared to the 12h-fasting (red) and 24h-fasting (green) conditions. Patterns found in the fed condition are used as reference. **c** Pseudospatial trajectories and UMAP distribution of *Deptor* expression in the fed, 12h-fasting, and 24h-fasting conditions. **d** Validation by in situ hybridization of *Deptor* mRNA distribution along the 3V in the fed vs. 24h-fasting condition (scale bars = 100 μm). Hypothalamic nuclei were delimited using DAPI staining. **e** Quantification of spatial dorso→ventral shift in *Deptor* gene expression along the 3V during 24h-fasting. n = 6 mice for the fed group and n = 7 for the 24h-fasting

group in two different cohorts (two-way ANOVA; Bonferroni post-tests; p < 0.001 for Deptor staining, p < 0.001 for dorsal part). **f** Heatmaps of shared transcription factor activity (enrichment scores calculated by AUCell[46]) in the fed, 12h-fasting, and 24h-fasting conditions ordered according to the pseudospatial trajectory. **g** Tanglegrams showing the shift in transcriptional activity from the fed to 12h-fasting and the fed to 12h-fasting conditions. The first branching highlights three main clusters in the fed and 12h-fasting conditions (i.e., regulons primarily activated in ependymal, α1-α2-β1 and β2 regions), while four clusters are derived from the first branching at 24h-fasting. Lines connecting dendrograms' leaves across the metabolic conditions uncover many regulons changing their region of activity from β2 to α1-α2-β1 during fasting. In (**e**), data are means ± SEM. *p < 0.05, **p < 0.01, and ***p < 0.001 compared to the fed condition. See Supplementary Figs. 5, 6 and Data 4. Source data are provided as a Source Data file.

previously described[17,28]. Since tanycytes primarily express postsynaptic molecules, they would more likely interact with presynaptic parts and modulate neurotransmitter release. Further exciting studies are necessary for decoding such tanycyte-presynaptic terminal communications.

Energy imbalance affects tanycytes' gene expression and functions[8,18,27]. While previous studies reported changes in β-tanycytes, especially regarding their role in regulating food intake through blood-brain exchangers and metabolic sensors, our dataset revealed higher responsiveness during fasting for the continuum parenchymal tanycyte group. This disparity with the literature may rely on the fact that many of these studies focused on the vmARH and the ME (where β tanycytes are located)[17–19,23,24,27], whereas few studies characterize α tanycyte populations[55,56]. Additionally, previous analyses in gene expression dynamics were performed at the tissue level[27], or at best, on FACS-isolated tanycytes[18], limiting the characterization of gene expression dynamics in the different tanycyte subgroups. Indeed, our present results highlight the importance of considering tanycyte heterogeneity while studying changes in gene expression: an absence of difference in bulk-RNAseq does not necessarily reflect the absence of changes in a specific subpopulation or a shift between different subpopulations.

During fasting, differential gene expression on clusters revealed a functional reorganization in tanycyte populations. More specifically, protein processing shuts down, whereas mRNA processing increases. This increase could be related to mRNA stabilization to maintain some expression or to allow rapid protein translation during refeeding. Similar processes were recently described in NPY neurons in response to fasting[57]. Differential PSA (DPSA) also revealed that, during energy imbalance, tanycytes might interchange functions and phenotypes from one subgroup to another, especially within the continuum parenchymal tanycyte group, indicating that a dynamic classification would better reflect tanycyte heterogeneity. Our previous work already suggested such functional shifts along the 3V: indeed, dorsal β1-tanycytes can acquire organized tight junction proteins following 24h-fasting and contact newly fenestrated vessels in the vmARH, resembling β2 tanycytes[18]. Here, our DPSA detailed the substantial dorso→ventral shifts in genes related to lipid and cholesterol metabolism. So far, very few studies have addressed the role of tanycytes in lipid incorporation and response[58–60], but this promising field is developing. In particular, β tanycytes contain lipid droplets[12] and metabolize palmitate to maintain body lipid homeostasis[58]. Interestingly, obese mice present alterations in tanycyte lipid metabolism and lipid droplet contents[60]. Because lipids act as signaling molecules and can promote inflammation under a high-fat diet[61,62], our findings call for more studies to elucidate their role in tanycyte functions. Our DPSA also detailed the dorso→ventral shifts in genes related to tanycyte-neuron and tanycyte-synapse interactions. Associated with changes in tanycyte functions during fasting, this shift suggests an adaptive modulation of neuronal functions according to the metabolic state.

Conversely, neurons contacting tanycytes through synaptoid contacts[15,55] may also differently modulate tanycyte subgroup functions during fasting.

Altogether, revisiting and defining alternative scRNAseq analytic approaches allowed us to describe the heterogeneity and plasticity of tanycyte populations. Many factors, including changes in transcription factor activity and cell-cell communications, contribute to the reorganization of tanycyte gene expression and consequently functions through different metabolic conditions, urging us to apprehend the tanycytes as a complex and adaptive continuum population.

## Methods

### Ethical statement
Given that this study utilizes rodents, ethical approval was obtained from the Veterinary Office of Canton de Vaud (VD3634). All procedures were conducted under ethical principles and guidelines for research involving animals.

### Mice
2-to-4-month-old male Rosa26-floxed stop tdTomato mice (initially obtained from Jax) and C57Bl6/J mice (initially obtained from Charles River) were used in this study. Animals were housed in groups (from 2 to 5 mice per cage) and maintained in a temperature-controlled room (at 22−23 °C) on a 12h light/dark cycle with ad libitum access to a chow diet (Diet 3436; Provimi Kliba AG, Kaiseraugst, Switzerland). For genotyping, biopsies were collected, DNA extraction was performed using the HOTSHOT method, and PCR amplification was performed using KAPA2G ReadyMix Kit (Sigma Aldrich, KK5103) following the manufacturer's instructions. Primers used for PCR are available from the authors. All animal procedures were approved by the Veterinary Office of Canton de Vaud (VD3634) and performed at the University of Lausanne. The number of animals used for each experiment is present in the "quantifications" section, the figure caption, Supplementary data 1, and source data file.

### TdTomato expression in tanycytes
To induce tdTomato expression along the 3V, TAT-CRE fusion protein (Excellgen, EG-1001) was stereotactically infused into the 3V (500 nl to 2 μl over 3 min at 2 mg/ml; at the coordinates form the bregma of AP = −1.7 mm; ML = 0 mm; DV = −5.3 mm from cortex surface) of ketamine/xylazine-anesthetized mice (100 mg/kg and 20 mg/kg, respectively) one week before experiments. The quality of the injections was arbitrarily estimated by visualizing the backflow of a CSF drop after the needle removal (Supplementary Data 1a).

### Tissue dissociation and cell collection for single-cell RNA sequencing
Experiments were performed in fed (n = 5 mice), 12h-fasting (n = 5 mice), and 24h-fasting (n = 6 mice) conditions one week following TdTomato induction in the ependymal layer (Supplementary Data 1a).

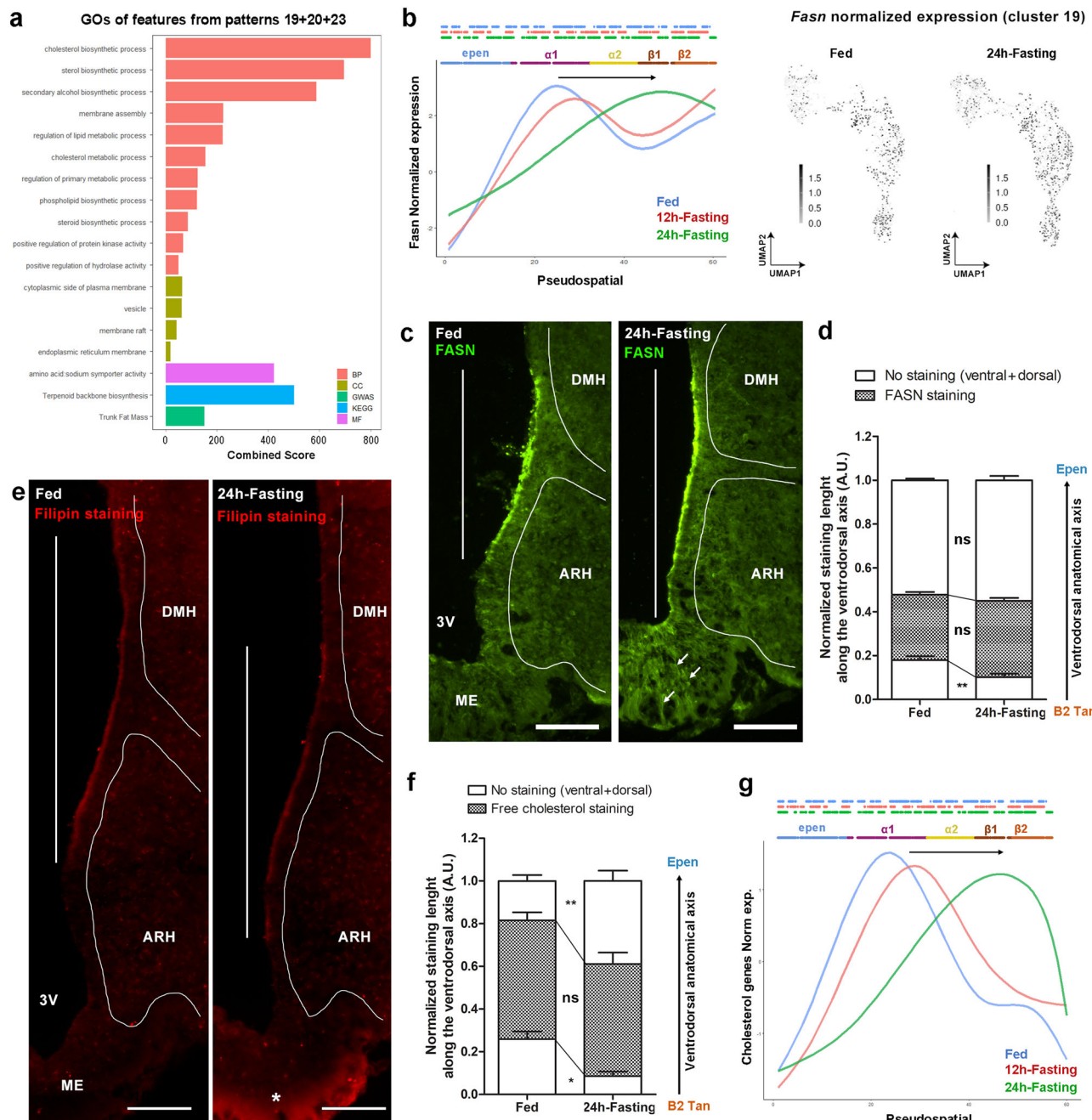

**Fig. 6 | Changes in lipid metabolism defined by differential pseudospatial analysis. a** Pattern characterization with Gene Ontology (GO) enrichment analysis highlighting biological process (BP), molecular function (MF), cellular component (CC), KEGG, and GWAS for the pseudospatial patterns displaying a dorso→ventral shift during 24h-fasting (pattern 19 + 20 + 23). The top terms are displayed and classified by Combined Score. **b** Pseudospatial trajectories and UMAP distribution of *Fasn* expression in the fed, 12h-fasting, and 24h-fasting conditions. **c** Validation by immunohistochemistry of FASN protein distribution along the 3V in the fed vs. 24h-fasting condition (scale bars = 100 μm). Hypothalamic nuclei were delimited using DAPI staining. Arrows indicate FASN staining in tanycyte processes. **d** Quantification of dorso→ventral shift in FASN protein distribution along the 3V during the 24h-fasting condition. n = 6 male mice for the fed group and n = 8 for the 24h-fasting group in 2 different cohorts (two-way ANOVA; Bonferroni post-tests; p < 0.01 for ventral part). Visualization (**e**) and quantification (**f**) of free cholesterol distribution using Filipin staining along the 3V in the fed vs. 24h-fasting condition (scale bars = 100 μm). Hypothalamic nuclei were delimited using Nissl staining. The asterisk indicates filipin staining in the median eminence. n = 6 male mice for the fed group and n = 6 for the 24h-fasting group in 2 different cohorts (two-way ANOVA; Bonferroni post-tests; p < 0.05 for the ventral part, p < 0.01 for the dorsal part). **g** Pseudospatial trajectories of genes involved in GO_cholesterol biosynthetic process in the fed, 12h-fasting, and 24h-fasting conditions. In (**d**, **f**), data are means ± SEM. *p < 0.05, **p < 0.01, and ***p < 0.001 compared to the fed condition. See Supplementary Fig. 6 and Data 4. Source data are provided as a Source Data file.

Mice had access to water ad libitum during fasting. Mice were killed between 8 a.m. and 9 a.m. MBH were microdissected using a binocular microscope and put in 500 μl ice-cold papain solution (20 μg/ml). Cells were then dissociated following Worthington's instructions. Briefly, cells were dissociated by incubating papain solution containing microdissected tissue at 37 °C for 30 min, followed by gentle manual trituration. Cell suspensions were centrifuged at 330 g for 5 min, and the cell pellet was then resuspended in a 500 μl albumin/ovomucoid protease inhibitor solution (1 mg/ml). A discontinuous density gradient centrifugation was performed by layering the cell suspension on

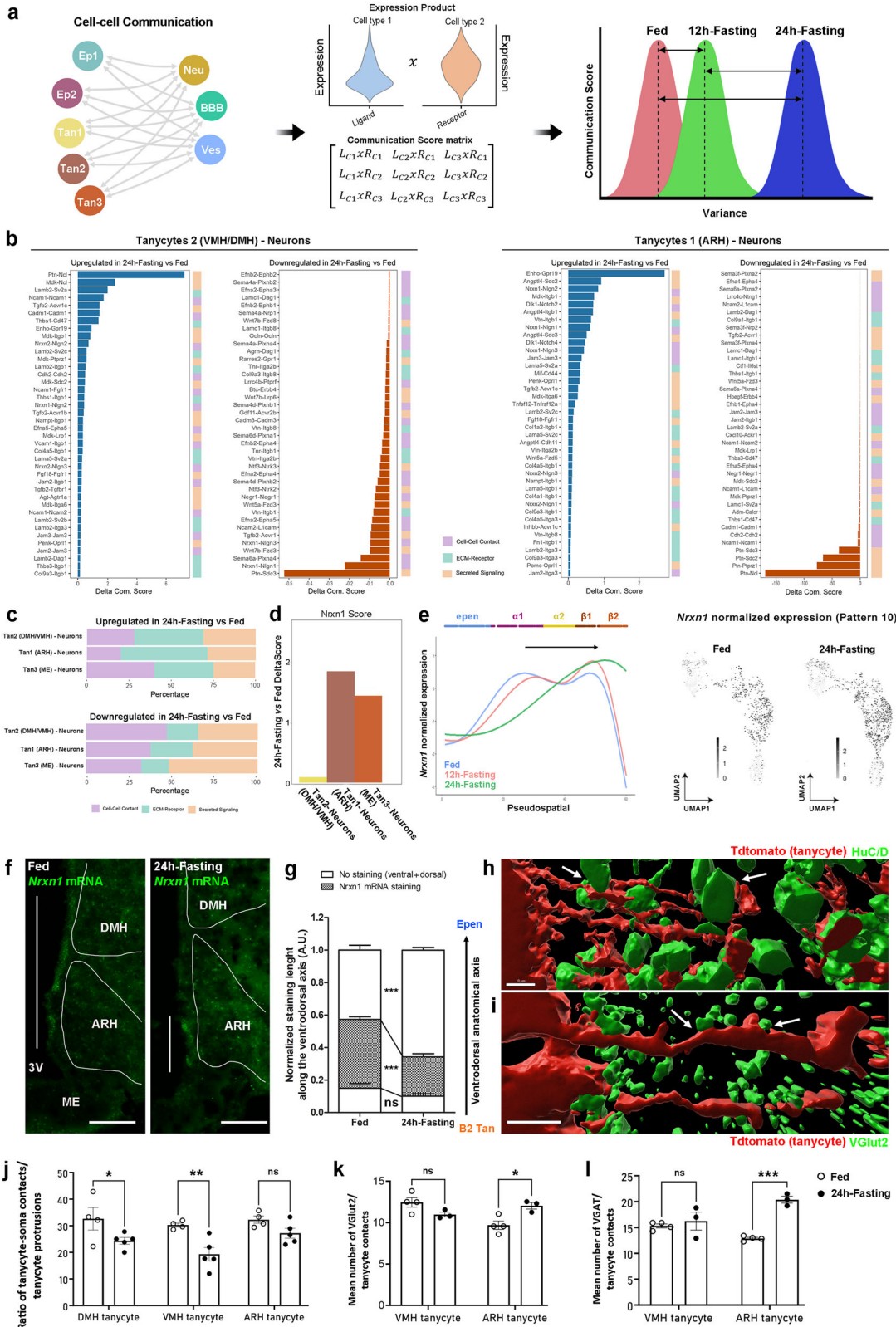

an 800 μl albumin/ovomucoid protease inhibitor solution (10 mg/ml) and centrifuged at 100 g for 7 min. Cell pellets were finally resuspended in 400 μl ice-cold calcium-free and magnesium-free HBSS.

TdTomato-positive singlet cells were sorted using a Beckman Coulter Moflo Astrios FAC-sorter according to their Forward and Side scattering properties (FSC and SSC), their negativity for DAPI (Viability dye, Blue DNA intercalating agent, ThermoFisher, cat nb D1306,

λEx/λEm (with DNA) = 358/461 nm), their positivity for RedDot1 (Viability dye, Far-red DNA intercalating agent, Biotium, #40060-1, λEx/λEm (with DNA) = 662/694 nm), and their level of tdTomato fluorescence emission (λEx/λEm = 554/581 nm). Gating parameters were set to improve the purity (i.e., fast collection, Noozle 70, purity) while keeping a loose FACS gate. Cells from identical conditions were collected and pooled in 10 μl in calcium-free and magnesium-free HBSS

**Fig. 7 | Changes in tanycyte-neuron interactions characterized by differential pseudospatial analysis and cell-cell communication. a** Summary of the analytic workflow for cell-cell communication. **b** Histograms showing the top up- and downregulated interactions from tanycytes 2 (i.e., VMH-DMH tanycytes) and tanycytes 1 (i.e., ARH tanycytes) populations with neurons. **c** Total up- and down-regulated percentage of LR pairs involved in cell-cell contact, ECM-receptor, and secreted signaling from tanycytes to neurons. **d** Total DeltaScore of LR pairs involving *Nrxn1* in interactions from tanycytes to neurons in the fed vs. 24h-fasting conditions. During fasting, communications through NRXN1 mainly increase in tanycytes 1 (i.e., ARH tanycytes) and tanycytes 3 (i.e., ME tanycytes) populations. **e** Pseudospatial trajectories and UMAP distribution of *Nrxn1* expression in the fed, 12h-fasting, and 24h-fasting conditions. **f** Validation by in situ hybridization of *Nrxn1* mRNA distribution along the 3V in the fed vs. 24h-fasting condition (scale bars = 100 μm). **g** Quantification of spatial dorso→ventral shift in *Nrxn1* expression along the 3V during 24h-fasting. Hypothalamic nuclei were delimited using DAPI staining.

n = 6 male mice for the fed group and n = 7 for the 24h-fasting group in 2 different cohorts (two-way ANOVA; Bonferroni post-tests; p < 0.001 for the staining, p < 0.001 for the dorsal part). 3D representation of a tanycyte/soma (**h**) and tany-cyte/synapse (**i**) interactions (arrows) built from immunohistochemistry images (scale bars = 10 μm). Number of tanycyte/soma (**j**), tanycyte/glutamatergic synapse (**k**), and tanycyte/GABAergic synapse (**l**) interactions in the VMH vs. the ARH according to the metabolic conditions. For tanycyte/soma interactions, n = 4 male mice for the fed group and n = 5 for the 24h-fasting group in one cohort (two-way ANOVA; Bonferroni post-tests; p < 0.05 for the DMH, p < 0.01 for the VMH). For tanycyte/synapse interactions, n = 4 male mice for the fed group and n = 3 for the 24h-fasting group in one cohort (two-way ANOVA; Bonferroni post-tests; p < 0.05 for the VMH for glutamatergic synapses; p < 0.001 for the ARH for GABAergic synapses). In (**g, j, k,** and **l**), data are means ± SEM. *p < 0.05, **p < 0.01, and ***p < 0.001 compared to the fed condition. See Supplementary Fig. 6 and Data 4−6. Source data are provided as a Source Data file.

(Merck, H8264) with 50% FBS (Gibco, 16030074), yielding a 500−600 cells/μl suspension (Supplementary Data 1a).

**Single-cell capture, cDNA library preparation, and sequencing**
Sorted cells were accurately counted using a hemocytometer with Trypan blue staining to validate the viability (at least 80%). A Chromium Next GEM Chip B (10X Genomics) was loaded with approximately 5000 cells, and sequencing libraries were prepared strictly following the manufacturer's recommendations (manual CG000183 revA). Briefly, an emulsion encapsulating single cells, reverse transcription reagents, and cell barcoding oligonucleotides were generated. After the reverse transcription step, the emulsion was broken, and double-stranded cDNA was generated and amplified for 12 cycles in a bulk reaction. This cDNA was fragmented, a P7 sequencing adaptor-ligated, and a 3' gene expression library generated by PCR amplification for 14 cycles.

Libraries were quantified using a fluorimetric method, and their quality was assessed on a Fragment Analyzer (Agilent Technologies). Cluster generation was performed with 140 pM of an equimolar pool from the resulting libraries using the Illumina HiSeq 3000/4000 PE Cluster Kit reagents. Sequencing was performed on the Illumina HiSeq 4000 using HiSeq 3000/4000 SBS kit reagents. Sequencing data were demultiplexed using the bcl2fastq2 Conversion Software (v. 2.20, Illumina), and primary data analysis was performed with the Cell Ranger Gene Expression pipeline (version 3.1.0, 10X Genomics).

**Individual sample processing.** Data processing was implemented similarly for the three conditions (i.e., fed, 12h-fasting, and 24h-fasting) using R (version 4.1.2). Quality control and preprocessing were performed using SingleCellExperiment (version 1.14.1) to construct the object with barcodes, features, and unique molecular identifiers (UMI) matrix. To retrieve a maximum number of high-quality cells, empty-Drops (dropletUtils version 1.12.3) was applied with default parameters to remove empty barcodes. Additionally, the percentage of reads mapping to the mitochondrial genome is usually used as a proxy to remove dead/dying cells. However, a recent publication[63] demonstrated that the fraction of mitochondrial reads depends on the tissue of origin, cell type, and experimental conditions: the usual strict cut-off of 5% could eliminate a large proportion of "valid" cells. As these variations may occur in our experimental design (i.e., fed vs. fasted conditions), isOutlier used with default parameters (scuttle version 1.2.1) was preferred to arbitrary cut-offs. Briefly, a cell is removed if the value of the mitochondrial expression is 3 MADs (median absolute deviation, the default) away from the median of expression in the population. Finally, to detect and remove putative doublet cells, scDblFinder (version 1.6.0) was employed, and cells with a doublet score >5 (arbitrary) were removed from the analysis. Before data normalization with Seurat (version 4.0.6), a last quality control step was added to remove all cells with less than 200 genes detected.

**Integrated analysis**
All single-cell datasets were integrated together[64] to compare clusters between the different metabolic conditions (Fed, 12h-Fasting, and 24h-Fasting) in a common background. Quality control and data normalization were done independently for each sample with the same parameters as above. For the integration, datasets are processed with SelectIntegrationFeatures(), PrepSCTIntegration(), FindIntegrationAnchors(), and IntegrateData() with default parameters. These functions find variable features common between all datasets, identify anchors (pair of cells from each dataset determined as a mutual nearest neighbor (MNN)), and integrate datasets based on filtered anchors[64]. The further processing is the same as the individual sample processing.

**Clustering analysis**
Primary processing was done with Seurat using quality-controlled cells in the fed, 12h-fasting, 24h-fasting conditions, and integrated dataset as input. Data normalization was performed with SCTransform() followed by Principal Component decomposition (RunPCA(), default parameter) and non-linear dimension reduction (RunUMAP(), with n_dim dimensions, where n_dim is selected as follows: the number of dimensions corresponding to the 3rd quantile of the explained variance distribution rather than being set arbitrarily based on the visual inspection of ElbowPlot()). Clustering was performed from the UMAP representation by calculating a Share Nearest Neighbor graph (FindNeighbors()0) and applying a modularity-based cluster detection FindClusters() with default parameters.

Cell-type assignment per cluster was done by i. extracting markers (FindMarkers()) and ii. comparing them with current knowledge and literature revision. To identify the biological processes characteristics for each cluster, enrichment analysis was performed with enrichR (version 3.1) using the most recent databases for "Biological Process (2021)", "Cellular Component (2021)", "KEGG (2019)", and "GWAS Catalog 2019".

**Supervised pseudospatial analysis (PSA)**
To identify the pseudospatial gradual patterns along the 3V, normalization, dimensional reduction, and clustering were performed in the feeding condition as described above. Note that in our dataset, ependyma cell populations are clustered in an anatomically-like distribution. The resulting Seurat object was converted to CellDataSet using the *as.cell_data_set* function of *Seurat-wrapper*. Following conversion, we used Monocle3's functions *learn_graph()* and *order_cells()* with default parameters to order cells based on their pseudospatial trajectory from ependyma to β2 tanycytes. To identify genes highly expressed to each ependyma cell subgroup as specific markers versus genes spanning several populations as shared markers, we generated the function PGEA(), allowing us to calculate the correlation between each gene expression vector and binary vectors representing simple or

combined populations (1 if the cell belongs to the population, 0 otherwise). Correlations were performed using the *rcorr* function (Hmisc v. 5.01). Only significant (p.adjusted < 0.01) and positive correlations were considered. Then, for each gene, a score in the form of the product of the correlation and its weighted average expression was calculated to normalize gene expression levels. A gene was classified as specific to a given ependyma population or belonging to a combined population according to the highest score among all possible classifications. Pseudospatial trajectories were visualized using a cubic spline-based regression on pseudospatial gradients and expression levels. Similar processing was done for the 12h- and 24h-fasting conditions.

### Unsupervised pseudospatial analysis (PSA)
The integrated Seurat object with the three metabolic conditions was used to perform unsupervised pseudospatial analysis. Using Tradeseq default parameters, pseudospatial gene expression patterns were calculated separately for the fed, 12h-, and 24h-fasting conditions. The number of patterns was subsetted to those with more than 50 genes. Next, we leveraged the "Tradeseq cluster analysis" to infer transition regions along the pseudospatial trajectory, allowing us to determine switches in gene expression patterns. To do so, we generated a binary matrix for each Tradeseq pseudospatial pattern, where pseudospatial points are set to 1 when an on-off switch in gene expression occurs and 0 for the rest. On-off switches in gene expression were determined using different thresholds (i.e., normalized expression at -/+0.1 to -/+0.5). Then, the binary matrix of the different patterns was summed to obtain transition scores for each pseudospatial point: the higher scores indicate regions with numerous on-off switches in gene expression.

### MERFISH analysis
Section 38 of the C57BL6J-638850 dataset (C57BL6J-638850.38) on the Allen Brain Atlas whole-brain spatial transcriptomics was used to validate gene expression patterns along the third ventricle (https://alleninstitute.github.io/abc_atlas_access/descriptions/MERFISH-C57BL6J-638850.html). IDs of 3V cells were extracted from the metadata table subsetted to cell types {"321 Astroependymal NN", "322 Tanycyte NN", "323 Ependymal NN"} with coordinates (y > 7 and 5.35 < x < 5.7). The data were represented as a scatter plot using matplotlib version 3.8.0 and smoothed with UnivariateSpline (scipy.interpolate.package, version 1.11.4).

### Differential gene expression analysis (DGEA)
The integrated Seurat object with the three metabolic conditions was used to perform differential gene expression analysis. Default Seurat workflow was questioned by a recent publication[65] showing many false positive differentially expressed genes influencing downstream analyses. Thus, we designed an alternate workflow based on pseudobulks, taking advantage of the proven reliability of the DESeq2 package (version 1.38.3). Briefly, each dataset (i.e., the three conditions) was split into four pseudo-replicates with an equivalent number of cells per cluster, and DESeq2 with default parameter was applied.

### Differential pseudospatial analysis (DPSA)
The integrated Seurat object (with normalized and integrated counts) was used to determine differences in pseudospatial gene expression gradients along the ventricle across metabolic conditions. Using Tradeseq, we first determined the trajectories of the fed pseudospatial patterns in the 12h- and 24h-fasting conditions. Finally, we performed a correlation-based analysis to compare the trajectories according to the metabolic conditions. Differences were set according to *p*-values corresponding to significant level of correlations.

### Transcription factors activity estimation
We used pySCENIC v.0.12.1 to process the transcriptional profile of all known transcription factors in the three metabolic conditions. Transcription factors activity has been estimated using the AUCell module from the regulons obtained from the standard GNRBoost processing of the co-expression matrix. The heatmap has been generated by clustering on the columns (Genes) only (parameter Rowv = NA) to keep the pseudospatial ordering on the rows.

### Inference of tanycyte-neuron communications
To infer mechanisms of cell-cell communication between tanycytes and neighboring subpopulations in the scRNAseq, we generated a method based on the expression product of ligand-receptor interactions. To infer cell-cell interactions, we used the ligand-receptor pairs database from CellChat[66]. Using the integrated Seurat object, we subset each condition (i.e., fed, 12h-fasting, and 24h-fasting). For each ligand-receptor pair, a function calculates Communication Scores based on the average of the product of expression from each cell of one cluster (expressing the ligand) to each of the other clusters (expressing the receptor). An ANOVA was calculated on the matrix of communication scores for a given ligand-receptor pair on the interaction between two cell types to determine the differences between conditions.

### Tissue collection
For immunohistochemistry and in situ hybridization, C57Bl6 and TAT-Cre-injected tdTomato mice were anesthetized with isoflurane and perfused transcardially with 0.9% saline, followed by an ice-cold solution of 4% paraformaldehyde (Applichem, A3813) in 0.1 M phosphate buffer (pH 7.4). Brains were quickly removed, postfixed in the same fixative for two hours at 4 °C, and immersed in 20% sucrose in 0.1 M phosphate-buffered saline (PBS) at 4 °C overnight. Brains were finally embedded in an ice-cold OCT medium (optimal cutting temperature embedding medium, Tissue Tek, Sakura, 4583) and frozen on liquid nitrogen-cooled isopentane. Brains were cut using a cryostat into 20-µm-thick coronal sections.

For electron microscopy, mice were anesthetized with isoflurane and perfused transcardially with 0.9% saline, followed by an ice-cold solution of 2% paraformaldehyde/2% glutaraldehyde in 0.1 M phosphate buffer, pH 7.4. Brains were quickly removed and postfixed in the same fixative overnight at 4 °C. 200 µm-thick hypothalamic slices were then cut using a vibratome. Afterward, the samples were incubated in 2% (wt/vol) osmium tetroxide and 1.5% (wt/vol) K4[Fe(CN)6] in 0.1 M PB buffer for 1 h, followed by one-hour incubation in 1% (wt/vol) tannic acid in 0.1 M PB buffer. Subsequently, brain slices were incubated in 1% (wt/vol) uranyl acetate for 1 h and dehydrated at the end of standard gradual dehydration cycles in ethanol. Samples were flat embedded in an Epon-Araldite mix[67,68]. All procedures were performed at room temperature. Next, polymerized flat blocks were trimmed using 90° diamond trim tool, and the arrays of 80 nm sections were obtained using 35° ATC diamond knife (Diatome, Biel, Switzerland) mounted on Leica UC6 microtome (Leica, Vienna). Sections were directly transferred to 2x4 cm pieces of silicon wafers using a modified array tomography procedure[15,69].

### Immunohistochemistry
20 µm slide-mounted sections were (1) blocked for 30 min using a solution containing 4% normal goat serum and 0.3% Triton X-100 (Sigma Aldrich, T8787) and (2) incubated overnight at 4 °C with primary antibodies followed by 2 h at room temperature with a cocktail of secondary Alexa Fluor-conjugated antibodies (1:500; Thermo Fisher). The primary antibodies used in this study are anti-FASN (1:500; Abcam, ab22759), anti-HuC/HuD (1:200; Invitrogen, A21271), anti-VGLUT2 (1:500; Synaptic system, 135403), and anti-VGAT (1:500; Synaptic system, 131004). Finally, the slide-mounted sections were (3) counterstained with DAPI (1:10,000; Sigma Aldrich, 10236276001) and (4) coverslipped using Mowiol (Sigma Aldrich, 81381). Images were acquired on fixed parameters using Nikon Eclipse 90i (x20 objective)

for FASN staining and Leica Thunder Imaging System (x63 objective, 0.3 μm z-scan, 12−13 optical slices) for VGAT and VGLUT2 stainings. Images for illustration were finally exported in.tiff. for the processing steps (i.e., adjust brightness and contrast, change colors, and merge images) using Adobe Photoshop (Adobe Systems, San Jose, CA).

## In situ hybridization

20 μm slide-mounted sections were processed for RNAscope® in situ hybridization following the manufacturer's instructions (ACD). Briefly, slide-mounted sections were first (1) incubated in a boiling 1X Target retrieval solution for 5 min, and (2) incubated at 40 °C with Protease III solution for 20 min. (3) The sequential hybridizations were then performed following the manufacturer's instructions. The probes used in this study are *Deptor* (C1-481561) and *Nrxn1* (C3-461511). (4) Slide-mounted sections were counterstained with DAPI and coverslipped using Mowiol. Images were acquired on fixed parameters using Nikon Eclipse 90i (x20 objective).

## Filipin staining

Fixed 20-μm-thick slide-mounted sections were (1) stained 30 min in filipin solution (0.05 mg/ml; Merck, SAE0087), (2) stained 20 min with NeuroTrace 640/660 Nissl stain (1/150; ThermoFisher, N21483), and (3) coverslipped using Mowiol. Images were acquired on fixed parameters using Nikon Eclipse 90i (x20 objective).

## Electron microscopy

Wafers were analyzed using FEI Helios Nanolab 650 scanning electron microscope (Thermo Fischer, Eindhoven). The imaging settings were as follows: MD detector, accelerating voltage 2 kV, current 0.8 nA, dwell time 4−6 μs. Images were collected manually or using the AT module of MAPs program (Thermo Fischer, Eindhoven). Single images were aligned and 3d-reconstructed with the IMOD software package (Kremer et al., 1996). For electron microscopy data interpretation, previous reports in the literature were used to recognize the different neural cell types based on their ultrastructural characteristics[15,70].

## Quantifications

For fluorescent microscopy *(Deptor, Nrnxn1*, filipin, and FASN staining), images were processed (threshold) and quantified using ImageJ. The length of the staining (or an absence of staining) was reported on the ventricle size. This ratio was calculated on a minimum of 4 pictures on the entire anteroposterior axis to obtain a mean value per mouse. N = 6 fed and n = 7 fasted mice were used for *Deptor* and *Nrxn1* mRNA staining. N = 6 fed and n = 8 fasted mice were used for FASN protein staining. N = 6 fed and n = 6 fasted mice were used for filipin staining.

For tanycyte-synapse interactions using electron microscopy, 3d-reconstructed images from 7 different tanycytes in 4 different mice were considered. The proportion of contact with pre-terminal, post-terminal, symmetric, or asymmetric synapses per tanycyte was calculated among all observed tanycyte-synapse interactions.

For tanycyte-synapse interactions using immunohistochemistry, images were loaded in Imaris x64 9.0.2 (Bitplane) to allow 3d reconstruction of synapses ending on tanycyte processes. tdTomato expressing tanycytes and VGLUT2+ and VGAT+ presynaptic terminals were reconstructed using 'Surface'. The number of contacts was quantified manually along tanycyte proximal processes (on 190 μm). This quantification was done on 32 tanycytes per nucleus on the entire anteroposterior axis to obtain a mean value per mouse. N = 4 fed and n = 3 fasted mice were used for this analysis.

For tanycyte-neuron interactions, we first counted the number of tanycyte protrusions (e.g., swelling and boutons) in the region of interest and then the number of these protrusions in contact with HuC/D-positive neurons. The regions of interest (i.e., ARH, VMH, and DMH) were identified based on DAPI staining. The analysis was performed on 4 sections on the anteroposterior axis to obtain a mean

value per mouse. N = 4 fed and n = 5 fasted mice were used for this analysis.

## Statistics and reproducibility

All values are expressed as means ± SEM. Data were analyzed for statistical significance with Graph Prism 5 software (Version 11.0), using t-test, one-way ANOVA followed by a Tukey's post-hoc test, or two-way ANOVA followed by a Bonferroni's post-hoc test when appropriate (see in the figure captions). The number of animals used for each experiment is given in the "quantifications" section and in the figure caption. *P*-values of less than 0.05 were considered statistically significant. No statistical method was used to predetermine sample size. No data were excluded from the analyses. The investigators were not blinded to allocation during experiments and outcome assessment.

## Reporting summary

Further information on research design is available in the Nature Portfolio Reporting Summary linked to this article.

## Data availability

The original single-cell transcriptomes reported in this paper have been deposited in the Gene Expression Omnibus (GEO) database under accession code GSE266664. The Hypomap dataset[43] was also used and is publicly available at https://www.repository.cam.ac.uk/items/8f9c3683-29fd-44f3-aad5-7acf5e963a75. The data supporting the findings of this study are also available from the corresponding authors upon request. Source data are provided with this paper.

## Code availability

R codes are available through the GitHub repositories (https://github.com/dalodriguez/DPGEA & https://github.com/dalodriguez/ExPCom). A processed single-cell dataset is available in these repositories to replicate our results. Code used for the remaining analysis was done using well known tools using standard workflows fully described in the methods section.

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

## Acknowledgements

Encapsulation, library preparation, and sequencing were performed at the Lausanne Genomic Technologies Facility, University of Lausanne, Switzerland (https://www.unil.ch/gtf/en/home.html). The authors also thank the CIG mouse facility (UNIL), the mouse Metabolic Evaluation Facility (MEF-UNIL), and the electron microscopy facility (EMF-UNIL). This work was supported by the Swiss National Science Foundation (PZOOP3_174120) and the European Research Council Starting Grant (TANGO, No. 948196). M.B. and F.S. are supported by the Swiss National Science Foundation (310030_185292), Horizon2020 (847941), and Novartis Foundation for medical-biological research (18A052). A.M. is supported by the Swiss National Science Foundation (310030_205068). F.L. is supported by the European Research Council Starting Grant (TANGO, No. 948196), the Novartis Foundation for medical-biological research (18A040), and the Swiss National Science Foundation (PCEFP3_194551). B.T. received support from the European Research Council Advanced Grant (INTEGRATE, No. 694798) and the Swiss National Science Foundation (310030_182496).

## Author contributions

M.B., D.L.R., A.M. and F.S. analyzed the scRNAseq dataset and produced the figures. J.E.M., R.D., A.R., and T.D. performed experiments. F.S. coordinated the bioinformatic analysis. F.L. designed the study, performed experiments, analyzed neuroanatomical data, produced the figures, and wrote the manuscript. B.T. provided logistic and scientific support for the scRNAseq experiment. F.L. and F.S. oversaw the research. All authors contributed to the manuscript.

## Competing interests

The authors declare no competing interests.
