## [Peer Review File · Nature Communications]

REVIEWER COMMENTS

Reviewer #1 (Remarks to the Author):

In this manuscript, Brunner and colleagues use single cell RNAseq to characterise the different populations of hypothalamic ependymal cells in feeding and fasting. They then validate these findings with a mix of histology, immunohistochemistry and RNAscope. Very little is known about these cells, particular in dynamic situations, thus this resource will be well received by the broader neuroscience community.

The authors begin by using standard single cell analysis, but then based on their observations that many of the cell types do not clearly segregate, develop a novel 'pseudo-spatial' technique (which looks to be related to pseudo-time analysis) for a more nuanced approach to the data. They then subject the animals to a 12 and 24 hour fast, and examine the effects of this physiological challenge.

This is a clearly written and wonderful piece of work. I would only ask the authors two things. First, to be crystal clear that location on a UMAP plot does not equate to their location in the intact tissue (I am not suggesting the authors don't know this, but many readers won't, and its best to be clear with messaging). Second, clearly state how many of each cell type are in the feeding and fasted states.

Otherwise, a classy piece of work, producing a badly needed resource, that needs to be seen by the broader community.

Reviewer #2 (Remarks to the Author):

The manuscript discusses the exploration of the heterogeneity and plasticity of the third ventricle (3V) in adult mice using fluorescence-activated cell sorting (FACS)-associated single-cell RNA sequencing (scRNAseq). The authors use a combination of classical clustering analysis and novel pseudospacial trajectory analyses to characterize cell populations along the ventricle. The key findings include the identification of specific markers differentiating ependymal cell subgroups and the dynamic relocalization of functions in response to metabolic changes. The manuscript discusses the impact of energy imbalance on tanycytes' gene expression and functions,

emphasizing the importance of considering tanycyte heterogeneity during gene expression analysis.

Overall, the manuscript provides valuable insights into the heterogeneity and plasticity of tanycyte populations in the 3V, offering a nuanced understanding of their functions in response to metabolic changes. Key points of interest are the demonstration of significant state-dependent transcriptional plasticity in tanycyte gene expression which requires a revision of the standard classification schema used to distinguish tanycyte subtypes. This is of considerable general interest to investigators working in hypothalamic biology.

General Observations:

1. The results section provides a thorough exploration of tanycyte heterogeneity and plasticity in various metabolic states.

2. While there is extensive discussion of differential expression of genes along the dorso-ventral axis, there is no discussion of differential expression along the antero-posterior axis, despite previously published reports that occurs in tanycytes. Can the authors comment on this, or better yet, perform additional analysis to identify these genes?

3. Figure 2F - In Situ Hybridization and Allen MERFISH Spatial Data:

a. Suggest incorporating more recent Allen MERFISH spatial transcriptomics datasets for visualizing gene expression along the 3V.

b. Integration of recently-generated Allen MERFISH data could reveal additional markers and tanycyte subsets. (data is accessible here, <https://allen-brain-cell-atlas.s3.us-west-2.amazonaws.com/index.html>)

4. Figure 3:

a. Propose integrating FigS3F-G into Fig3 for better content organization.

b. Suggest moving Fig3B to FigS3 for improved figure organization.

5. Figure 4 - Gene Names Readability:

a. Note that gene names in Fig4F-G are barely readable (on paper or electronically). The authors should upload a high-resolution image for this figure.

General Comments:

- a. The work demonstrates a significant contribution to the field in understanding tanycyte dynamics during metabolic shifts, as well as provide key insights into how tanycytes (+ ependymal cells) respond to metabolic states to control hypothalamic functions.
- b. The methodology appears sound.
- c. Comparisons with established literature are well-incorporated, adding depth to the contextualization.
- d. The manuscript generally meets expected standards with minor revisions to figures.

Minor comments:

Line 340: Change “early” to “immediate-early”.

Conclusion: The work is original and significant, advancing our understanding of tanycyte and ependymal cells physiological response to internal metabolic states. While there are still some aspects in which the manuscript could be improved, it is overall well-positioned for publication.

Reviewer #3 (Remarks to the Author):

The manuscript discusses the exploration of the heterogeneity and plasticity of the third ventricle (3V) in adult mice using fluorescence-activated cell sorting (FACS)-associated single-cell RNA sequencing (scRNAseq). The authors use a combination of classical clustering analysis and novel pseudospacial trajectory analyses to characterize cell populations along the ventricle. The key findings include the identification of specific markers differentiating ependymal cell subgroups and the dynamic relocalization of functions in response to metabolic changes. The manuscript discusses the impact of energy imbalance on tanycytes' gene expression and functions, emphasizing the importance of considering tanycyte heterogeneity during gene expression analysis.

Overall, the manuscript provides valuable insights into the heterogeneity and plasticity of tanycyte populations in the 3V, offering a nuanced understanding of their functions in response to metabolic changes.

General Observations:

1. The results section provides a thorough exploration of tanycyte heterogeneity and plasticity in various metabolic states.

2. Figure 2F - In Situ Hybridization and Allen Merfish Spatial Data:

a. Suggest incorporating more recent Allen Merfish spatial transcriptomics datasets for visualizing gene expression along the 3V.

b. Integration of Allen Merfish data could reveal additional markers and tanycyte subsets. (data is accessible here, <https://allen-brain-cell-atlas.s3.us-west-2.amazonaws.com/index.html>)

3. Figure 3:

a. Propose integrating FigS3F-G into Fig3 for better content organization.

b. Suggest moving Fig3B to FigS3 for improved figure organization.

4. Figure 4 - Gene Names Readability:

a. Note that gene names in Fig4F-G are barely readable (on paper or electronically). Authors should upload a high-resolution image for this figure.

General Comments:

a. The work demonstrates a significant contribution to the field in understanding tanycyte dynamics during metabolic shifts, as well as provide key insights into how tanycytes (+ ependymal cells) respond to metabolic states to control hypothalamic functions.

b. The methodology appears sound.

c. Comparisons with established literature are well-incorporated, adding depth to the contextualization.

d. The manuscript generally meets expected standards with minor revisions to figures.

Conclusion: The work is original and significant, advancing our understanding of tanycyte and ependymal cells physiological response to internal metabolic states. While there are minor suggestions for improvement, the manuscript is well-positioned for publication.

Reviewer #4 (Remarks to the Author):

Brunner and colleagues performed single-cell RNA sequencing on FACS-isolated ependymal cells from fed, 12h-fasted, and 24h-fasted adult male mice. They implemented a pseudospacial analysis for the improved characterization of tanycytes and found that fasting influences gene expression patterns in tanycytes along the 3V inducing metabolic and functional switch particularly in the context of lipid metabolism. They also explore tanycyte-neuron and tanycyte-synapse interactions.

While this manuscript presents valuable insights and is certainly intriguing, there is a concern that the authors may be overemphasizing their findings with statements such as 'First, using clustering analysis, we confirmed that our current tanycyte classification is inadequate as numerous gene markers are shared between subpopulations, and their heterogeneity further increases with a fasting time course' (lines 128-130). However, upon closer examination, their reclassification effort appears to be a refined version of Campbell's paper (PMID: 28166221), which already contained similar information, including fasting data. It would be advisable for the authors to consider toning down such statements in the introduction, results, and discussion sections to better reflect the novelty and extent of their contributions. On a positive note, the second part of the manuscript introduces genuinely novel and interesting information. To enhance the impact of the paper, it would be beneficial if the authors could incorporate spatial transcriptomics to validate the 'unique gene markers' they have identified.

Specific comments:

1. For the FACS isolation of ependymal cells, they induce tdTomato expression in tanycytes and validate the specificity of tdtomato expression along the 3V in Figure S1. However, out of 13,121 individual cells retrieved after single-cell RNA sequencing, only 5481 (41.8%) were ependymal cells. Unlike Fig S1, they retrieve most of the cell populations in the region, such as four clusters of neurons, vascular leptomeningeal cells, microglia, pericytes etc (Fig 1b). They eventually subclustered tanycytes and ependymocytes for the analysis, but the efficiency of Tat-Cre injections between experimental groups remains questionable making the idea of FACS-assisted "enrichment" less significant. The authors mention the limitation of this approach in the results section, but line 176 could be rephrased "Our FACS-associated scRNAseq approach aimed to get a high-resolution transcriptomic profile for the ependymocytes lining the MBH".
2. The authors identify two populations of ependymocytes (Epen) with common features and specific features (Krt15+/Col6a5+ for Epen1, and Barhl2+/Pitx2+ for Epen2). But as stated in Line 186, the differences in *Ascc1* and *Csrp2* in the two Epen clusters are not evident in Fig 1e. Then Fig S2A shows a different set of Epen features with supporting ISH images from Allen Mouse Brain Atlas, but chosen ISH images are poor quality. The expression patterns are unclear, especially for the genes *C1qtnf3*, *Htr5b*, *Otx2*.
3. Campbell et al. (PMID: 28166221) already reported the presence of two ependymocyte clusters. How are these two Epen clusters different from the ones already reported?
4. Line 191 the figure reference should be Figure 1F (instead of Figure 1E).

5. Using pseudospacial analysis, they find specific and overlapping genes for each ependymal sub group. The authors must discuss more clearly how their workflow is original (Line 47) as established Monocle and Tradeseq workflows were used for pseudospacial analysis (PMID: 30787437, <https://cole-trapnell-lab.github.io/monocle3/>, <https://github.com/statOmics/tradeSeq>, PMID: 32139671).

6. But in comparison with DE analysis, how different are these genes obtained based on the pseudospace trajectory (Fig 1D)? The authors could provide an elaborate comparison between the shared and the unique features of each cluster obtained through DE gene analysis vs pseudospace trajectory. Line 209 states that “DE gene analysis does not define ependymal subgroups” but the authors compare or display only top 2-4 features in Fig 1E, making it difficult to draw a conclusion on the limitation of DE approach.

7. The genes, S100b, Cdh2, Col25a1 represented in Fig 1E and Fig 1F are already known to be expressed in tanycytes (PMID: 36949050, PMID: 28166221). The authors didn’t find unique features for β 1 tanycyte although it is known that marker Sprr1a is specifically localized in β 1 tanycytes (PMID: 28166221).

8. Validating the localization of at least some of the unique unknown features identified in Fig 2D by fluorescent ISH would be necessary to strengthen the utility of the cell type classification method in the paper

Responses to reviewers' comments:

We thank the reviewers and the editor for their thoughtful and constructive comments on our manuscript. Each of these is addressed in a point-by-point manner. Moreover, by following the checklists and during the rereading process, a few changes were made. The sex was clearly indicated in the abstract and the text. Data and code accessibility were added to the methods. Missing information, notably in the legends and in methods, was clarified. A few minor errors such as typos, graph formatting, stat and data reporting, and unclear descriptions were also corrected.

REVIEWER COMMENTS

Reviewer #1 (Remarks to the Author):

In this manuscript, Brunner and colleagues use single cell RNAseq to characterise the different populations of hypothalamic ependymal cells in feeding and fasting. They then validate these findings with a mix of histology, immunohistochemistry and RNAscope. Very little is known about these cells, particular in dynamic situations, thus this resource will be well received by the broader neuroscience community.

The authors begin by using standard single cell analysis, but then based on their observations that many of the cell types do not clearly segregate, develop a novel 'pseudo-spatial' technique (which looks to be related to pseudo-time analysis) for a more nuanced approach to the data. They then subject the animals to a 12 and 24 hour fast and examine the effects of this physiological challenge.

This is a clearly written and wonderful piece of work. I would only ask the authors two things. First, to be crystal clear that location on a UMAP plot does not equate to their location in the intact tissue (I am not suggesting the authors don't know this, but many readers won't, and its best to be clear with messaging). Second, clearly state how many of each cell type are in the feeding and fasted states.

Otherwise, a classy piece of work, producing a badly needed resource, that needs to be seen by the broader community.

We thank the reviewer for her/his comments about our work. We added the term “mimic” in the results (line 222) and the sentence “Defined as “pseudo” as the UMAP plot does not represent exact cell location in the intact tissue” in the discussion (line 495-496) to clarify that the cell location on a UMAP plot is not their location in the intact tissue.

Moreover, the number of cells in each metabolic state and each cell type is available in Table S1. We also added the information in the main text (lines 138 and 141).

Reviewer #2 and #3 (Remarks to the Author):

The manuscript discusses the exploration of the heterogeneity and plasticity of the third ventricle (3V) in adult mice using fluorescence-activated cell sorting (FACS)-associated single-cell RNA sequencing (scRNAseq). The authors use a combination of classical clustering analysis and novel pseudospacial trajectory analyses to characterize cell populations along the ventricle. The key findings include the identification of specific markers differentiating ependymal cell subgroups and the dynamic relocalization of functions in response to metabolic changes. The manuscript discusses the impact of energy imbalance on tanycytes' gene expression and functions, emphasizing the importance of considering tanycyte heterogeneity during gene expression analysis.

Overall, the manuscript provides valuable insights into the heterogeneity and plasticity of tanycyte populations in the 3V, offering a nuanced understanding of their functions in response to metabolic changes. Key points of interest are the demonstration of significant state-dependent transcriptional plasticity in tanycyte gene expression which requires a revision of the standard classification schema used to distinguish tanycyte subtypes. This is of considerable general interest to investigators working in hypothalamic biology.

We thank the reviewer for her/his comments about our work.

General Observations:

1. The results section provides a thorough exploration of tanycyte heterogeneity and plasticity in various metabolic states.

2. While there is extensive discussion of differential expression of genes along the dorso-ventral axis, there is no discussion of differential expression along the antero-posterior axis, despite previously published reports that occurs in tanycytes. Can the authors comment on this, or better yet, perform additional analysis to identify these genes?

We found a few genes with an anteroposterior distribution, but we did not include the antero-posterior axis in our analysis due to technical/theoretical issues. First, the trajectories on the antero-posterior axis were less clearly defined on the UMAP distribution, unlikely the dorso-ventral axis. Moreover, as very little is known about gene expression on this axis, we had no way to robustly validate an eventual antero-posterior pseudospacial index. We added a few sentences about this in the discussion (lines 500-502).

3. Figure 2F - In Situ Hybridization and Allen MERFISH Spatial Data:

a. Suggest incorporating more recent Allen MERFISH spatial transcriptomics datasets for visualizing gene expression along the 3V.

b. Integration of recently generated Allen MERFISH data could reveal additional markers and tanycyte subsets. (data is accessible here, <https://allen-brain-cell-atlas.s3.us-west-2.amazonaws.com/index.html>)

We thank the reviewer for this advice. We looked at and analyzed the different Allen MERFISH spatial transcriptomics datasets. We used C57BL6J-638850 dataset (as the 3v is visible and the resolution is good in this dataset) to visualize and validate our pseudospacial analysis by analyzing gene expression along the 3V, notably *Serpine2* and *Gli3*. Similar trajectories were observed. However, these data also

reveal that the spatial (around 500 cells along the 3v) and/or transcriptomic (around 500 genes in the used dataset, absence of Npy mRNA for instance) resolution for such approach is still limited, one of the reasons we chose to develop the pseudospacial analysis.

We added figure S4 and a description in the results (lines 251-253) and the methods (lines 689-695).

4. Figure 3:

- a. Propose integrating FigS3F-G into Fig3 for better content organization.
- b. Suggest moving Fig3B to FigS3 for improved figure organization.

Changes have been made.

5. Figure 4 - Gene Names Readability:

- a. Note that gene names in Fig4F-G are barely readable (on paper or electronically). The authors should upload a high-resolution image for this figure.

Changes have been made.

General Comments:

- a. The work demonstrates a significant contribution to the field in understanding tanycyte dynamics during metabolic shifts, as well as provide key insights into how tanycytes (+ ependymal cells) respond to metabolic states to control hypothalamic functions.
- b. The methodology appears sound.
- c. Comparisons with established literature are well-incorporated, adding depth to the contextualization.
- d. The manuscript generally meets expected standards with minor revisions to figures.

Minor comments:

Line 340: Change "early" to "immediate-early".

Changes have been made.

Conclusion: The work is original and significant, advancing our understanding of tanycyte and ependymal cells physiological response to internal metabolic states. While there are still some aspects in which the manuscript could be improved, it is overall well-positioned for publication.

Reviewer #4 (Remarks to the Author):

Brunner and colleagues performed single-cell RNA sequencing on FACS-isolated ependymal cells from fed, 12h-fasted, and 24h-fasted adult male mice. They implemented a pseudospacial analysis for the improved characterization of tanycytes and found that fasting influences gene expression patterns in tanycytes along the 3V inducing metabolic and functional switch particularly in the context of lipid metabolism. They also explore tanycyte-neuron and tanycyte-synapse interactions.

While this manuscript presents valuable insights and is certainly intriguing, there is a concern that the authors may be overemphasizing their findings with statements such as 'First, using clustering analysis, we confirmed that our current tanycyte classification is inadequate as numerous gene markers are shared between subpopulations, and their heterogeneity further increases with a fasting time course' (lines 128-130). However, upon closer examination, their reclassification effort appears to be a refined version of Campbell's paper (PMID: 28166221), which already contained similar information, including fasting data. It would be advisable for the authors to consider toning down such statements in the introduction, results, and discussion sections to better reflect the novelty and extent of their contributions. On a positive note, the second part of the manuscript introduces genuinely novel and interesting information. To enhance the impact of the paper, it would be beneficial if the authors could incorporate spatial transcriptomics to validate the 'unique gene markers' they have identified.

We thank the reviewer for his/her advice on how to improve our work.

The clustering analysis at the beginning of the manuscript is not novel (and not described as such but defined as “standard” in comparison with the pseudospacial analysis we introduce later) and has been performed by others, notably by Campbell et al. We put this analysis at the beginning of our manuscript to first show that our results are consistent with what has been done before (known tanycyte subgroups, known markers, and known functions) but, also, to highlight its limitations. The novelty in our study is the supervised and unsupervised pseudospacial analysis, which lead to a revisited classification for tanycytes. While Campbell's clustering proposes six subgroups for tanycytes and 2 for ependymal cells, we conclude on only three main subgroups: 2 well-defined (beta and typical ependymal cells) and one « fuzzy » (transition or continuum tanycyte population) due to the gradual distribution and dynamics in gene expression. In contrast to Campbell et al., we based our classification on the spatial distribution of gene expression and notably the gradients in gene expression along the ventricle. To the best of our knowledge, such an approach was never undertaken before but needed for the community as previously discussed in various studies (PMID: 29351662, PMID: 31133987).

Its necessity is notably revealed when we compare the different metabolic conditions to apprehend tanycyte plasticity. Indeed, we revealed ventro-dorsal and dorso-ventral shifts in gene expression distribution along the ventricle, confirming that we should not consider different subgroups for tanycytes but rather a continuum of tanycytes that progressively change from the bottom to the top of the ventricle, and vice versa. Consequently, tanycyte functions may not be categorized by subgroups, but as global functional modules that move along the ventricle.

Another point of originality of our work is the detailed study of the impact of fasting on tanycyte heterogeneity and plasticity with different and complementary approaches. Indeed, even though the published Campbell dataset contains data regarding fasting, Campbell et al did not describe the dynamics of gene expression changes in tanycytes and the consequent modulatory effects on tanycytes' activity.

Therefore, we rephrased some sentences in the manuscript to highlight these aspects and better reflect the novelty and interest of our contributions to the scientific community.

Specific comments:

1. For the FACS isolation of ependymal cells, they induce tdTomato expression in tanycytes and validate the specificity of tdtomato expression along the 3V in Figure S1. However, out of 13,121 individual cells retrieved after single-cell RNA sequencing, only 5481 (41.8%) were ependymal cells. Unlike Fig S1, they retrieve most of the cell populations in the region, such as four clusters of neurons, vascular leptomeningeal cells, microglia, pericytes etc (Fig 1b). They eventually subclustered tanycytes and ependymocytes for the analysis, but the efficiency of Tat-Cre injections between experimental groups remains questionable making the idea of FACS-assisted "enrichment" less significant. The authors mention the limitation of this approach in the results section, but line 176 could be rephrased "Our FACS-associated scRNAseq approach aimed to get a high-resolution transcriptomic profile for the ependymocytes lining the MBH".

We aimed to enrich the cell suspension with ependymal cells (tanycyte + typical ependymal cells) to focus on their gene expression profile. We obtained a 42% enrichment. For comparison, Campbell's paper has 5155/20,921 (24%) ependymal cells in their dataset (considering 6 different metabolic and sexual states, compared to only three in our dataset). The efficiency of Tat-Cre injections between experimental groups turned out to be as we expected as they were made in 15 mice in basal fed condition one week before the metabolic challenge/sacrifice. As a proof, the number of ependymal cells collected in the three different metabolic conditions is similar, around 40%. Moreover, tdTomato expression is mainly in the ependyma (around 80% of tdTomato⁺ cells are ependymal cells). The limitation is likely due to the parameters used for cell sorting, notably a loose gating strategy to be sure to have enough cells for single-cell analysis: it is discussed in the manuscript at lines 157-160.

Line 176 has been rephrased as follow: " Our FACS-associated scRNAseq approach initially aimed to characterize transcriptomic profiles of the ependymocytes lining the MBH. We next subsetted the data to focus on tanycytes, typical ependymal cells, and astrocytes in the fed condition to reveal a finer heterogeneity potentially hidden in the initial clustering workflow (Figure 1D, Table S1D)." (lines 167-170).

2. The authors identify two populations of ependymocytes (Epen) with common features and specific features (Krt15+/Col6a5+ for Epen1, and Barhl2+/Pitx2+ for Epen2). But as stated in Line 186, the differences in Ascc1 and Csrp2 in the two Epen clusters are not evident in Fig 1e. Then Fig S2A shows a different set of Epen features with supporting ISH images from Allen Mouse Brain Atlas, but chosen ISH

images are poor quality. The expression patterns are unclear, especially for the genes *C1qtnf3*, *Htr5b*, *Otx2*.

Ascc1 and *Csrp2* expression distributions are not specific but enriched for Epen1 and Epend2, respectively, visible on the violin plot in fig1e with different gene expression profiles (but expressed in both populations). For the figure, we selected specific features (the list of “enriched” vs “specific” features is available in Table S2). A few of them were indeed unclear: therefore, we added arrows to highlight the staining and selected other images from Allen Brain atlas with a better quality (*i.e.*, *Car9* and *Fln7*) to improve the readability of the manuscript.

3. Campbell et al. (PMID: 28166221) already reported the presence of two ependymocyte clusters. How are these two Epen clusters different from the ones already reported?

We compared the two ependymocyte clusters found by Campbell et al. to ours. Generally, most of the markers were in common (around 60%) but our two ependymal populations do not match Campbell Ependymocyte classification. Indeed, most of the common markers (for both of our subgroups) were found with Campbell’s Epen1. This difference likely arises from the differences in the experimental protocol and the analytic approach. For instance, we used males in fed conditions only for this analysis, whereas Campbell et al used a mix of different metabolic and sexual conditions, thereby increasing the heterogeneity.

4. Line 191 the figure reference should be Figure 1F (instead of Figure 1E).

Changes have been made.

5. Using pseudospacial analysis, they find specific and overlapping genes for each ependymal subgroup. The authors must discuss more clearly how their workflow is original (Line 47) as established Monocle and TradeSeq workflows were used for pseudospacial analysis (PMID: 30787437, <https://cole-trapnell-lab.github.io/monocle3/>, <https://github.com/statOmics/tradeSeq>, PMID: 32139671).

We apologize if the term “original” in the abstract led to some misunderstanding. As specified by the reviewer and reported in the manuscript (both studies and algorithms mentioned by the reviewer have been used and cited in the manuscript), the concept of pseudotime as a mono-dimensional measure of the similarity of single cells in the UMAP were introduced in the literature before by Trapnell lab (Monocle) and others (Slingshot etc..). However, in the original interpretation, this concept has always been related to time (PMID: 30787437) and cell differentiation trajectories (PMID: 32139671, and many others). In our special and unique situation, the UMAP plot represented the ependymocytes according to their dorso-ventral axis (and then we validated this in Figure S1G-H, S2). This allowed us to use Monocle to order the cells on an anatomical (“pseudo-spatial”) trajectory and design a tailored gene gradient analysis as explained in *Methods - Pseudospacial analysis (PSA)*. The novelty here is the identification of specific *versus* shared features with respect to the pseudospacial trajectory based on the correlation between each gene expression vector and binary vectors representing simple or combined populations. We also inferred transition regions along the pseudospacial trajectory, allowing

us to determine switches in gene expression patterns, by building binary matrix based on gene expression.

We detailed this in the methods (lines 657-687) and decided to remove the term “original” from the abstract to avoid any future misinterpretation.

6. But in comparison with DE analysis, how different are these genes obtained based on the pseudospace trajectory (Fig 1D)? The authors could provide an elaborate comparison between the shared and the unique features of each cluster obtained through DE gene analysis vs pseudospace trajectory. Line 209 states that "DE gene analysis does not define ependymal subgroups" but the authors compare or display only top 2-4 features in Fig 1E, making it difficult to draw a conclusion on the limitation of DE approach.

We performed a comparison between markers found from the clustering analysis and those from the PS analysis. Basically, similar genes are also found in the PS (around 50% depending on the populations), as we are describing the same cell types (ependymal cells and tanycytes). However, the PS analysis allows to differentiate specific features *versus* shared features while cluster analysis only suggest specific markers, against the evidence of having several shared features for a1, a2 and b1 populations). As a consequence, the PS analysis is less "noisy", more restrictive with the identification of specific markers (cluster gives more genes) and allows the discovery of new markers (not revealed by cluster analysis) such as *Sema3f*, *Fign*, *Phldb2*, or *Npr1* (all validated by ISH). This comparison was added in figures S3 and table S3.

Regarding the limitation of clustering analysis, it is based on figure1E and supplementary table S2. In t figure1E, we displayed the top enriched (calculated by differential gene expression) and top “specific” (calculated using the $pc1$ (% of expressing cells in the cluster)/ $pc2$ (% of expressing cells elsewhere) ratio) features: 2-4 genes were displayed for panel readability, but the rest of the gene list is available in Table S2. With these top genes (in Fig1E), we are already able to appreciate that a1, a2 and b1 features are shared between different populations and are not specific: it would be even more evident if we were plotting the rest of the gene list.

7. The genes, *S100b*, *Cdh2*, *Col25a1* represented in Fig 1E and Fig 1F are already known to be expressed in tanycytes (PMID: 36949050, PMID: 28166221). The authors didn't find unique features for $\beta 1$ tanycyte although it is known that marker *Sprr1a* is specifically localized in $\beta 1$ tanycytes (PMID: 28166221).

As stated by the reviewer, these markers are known, and have been chosen to validate our PS analysis. The utility of this analysis here is to describe the spatial distribution of tanycytes' features, notably the known one. It enables their classification into specific or shared features, and, in that case, to which subgroups they overlap. Another utility of the PS analysis is also to find new markers: thus, we replaced (in fig.3e-f) and added (in fig. S3b-c) some genes to include these novel markers (based on the comparison done between the standard clustering analysis and the PS analysis (cf comment 6)).

Regarding *Sprr1a*, we did not find it neither with the standard clustering analysis in the fed condition nor with the pseudospacial analysis (in our dataset and the hypomap dataset). In our data, *Sprr1a* is expressed by a few b2 tanycytes and notably found to characterize this population in the fasted conditions. Similarly, even if a picture illustrates some specificity for beta1 in Campbell et al. (PMID:

28166221, fig.2e), in the same paper, *Sprr1a* mRNA is shown to be expressed in very few cells and mostly located in b2 tanycytes (PMID: 28166221, fig.2e). Moreover, *Sprr1a* is not present in the Campbell tanycyte feature list (PMID: 28166221, Supplementary Tables). Altogether, this data suggests this gene to be expressed at the b1/b2 boundary and, for all these reasons, we do not consider *Sprr1* as a marker for b1 tanycytes.

8. Validating the localization of at least some of the unique unknown features identified in Fig 2D by fluorescent ISH would be necessary to strengthen the utility of the cell type classification method in the paper

We first added new ISH pictures to show some unique unknown features emerged from the PS analysis (fig.3e-f and fig. S3b-c) (novel based on the comparison done between the standard clustering analysis and the PS analysis (cf comment 6)). We also incorporated more recent Allen MERFISH spatial transcriptomics datasets for visualizing gene expression along the 3V, notably *Serpine2* and *Gli3*, as further validation of the PS analysis (fig S4).

REVIEWER COMMENTS

Reviewer #1 (Remarks to the Author):

The authors have responded to all of my concerns.

Reviewer #2 (Remarks to the Author):

The authors have addressed all my outstanding concerns. Congratulations on a fine piece of work.

Reviewer #3 (Remarks to the Author):

I co-reviewed this manuscript with one of the reviewers who provided the listed reports. This is part of the Nature Communications initiative to facilitate training in peer review and to provide appropriate recognition for Early Career Researchers who co-review manuscripts

Reviewer #4 (Remarks to the Author):

The authors have addressed our concerns in the text of the manuscript distinguishing the standard workflows and the novelty of their work. They have made clear that similar gradual UMAP patterns were detected in mouse Hypomap too.

Additional comments:

The authors need to provide DAPI stained images for Fig 5d, Fig 6e, Fig 6c, Fig 7f for the clear visualization of 3V boundaries.

Fig6e: Fed Filipin staining: Is it the absence of staining or compromised 3V (close to ARH). Providing DAPI stained image would clarify this.

Fig 7f: Nr1h3 24h Fasting: Since the authors measure the length of staining along the ventrodorsal axis, it might be important to provide representative images with intact 3V.

Since part of the manuscript deals with tanycyte-to-neuron communication, the authors may want to discuss this recently published article on the topic: PMID: 38538787.

Response to reviewer

Additional comments:

We thank the reviewer for her/his comments to improve the quality of our pictures.

The authors need to provide DAPI-stained images for Fig 5d, Fig 6e, Fig 6c, and Fig 7f to clearly visualize 3V boundaries.

In the main figure, we used DAPI staining to draw 3V boundaries (white lines) in the ISH or IHC pictures. In supplementary figure 6, we added both DAPI + ISH or IHC images.

Fig6e: Fed Filipin staining: Is it the absence of staining or compromised 3V (close to ARH). Providing DAPI stained image would clarify this.

DAPI staining was not available for the Filipin staining as both were in the same channel. We produced new fed and 24h-fasted brains to perform a double Filipin and Nissl staining. As before, we used Nissl staining to draw 3V boundaries (white lines) in the main figure, and we added the new Nissl + Filipin pictures in supplementary figure 6.

Fig 7f: Nrnx1 24h Fasting: Since the authors measure the length of staining along the ventrodorsal axis, it might be important to provide representative images with intact 3V.

We replaced the picture with a clearer one.

Since part of the manuscript deals with tanycyte-to-neuron communication, the authors may want to discuss this recently published article on the topic: PMID: 38538787.

This publication was already added to the discussion for the first revision (as ref 55).

REVIEWERS' COMMENTS

Reviewer #4 (Remarks to the Author):

The authors have satisfactorily answered Reviewers' comments